# TRADING COVERAGE FOR PRECISION: CONFORMAL PREDICTION WITH LIMITED FALSE POSITIVES

## ABSTRACT

In this paper, we develop a new approach to conformal prediction in which we aim to output a precise set of promising prediction candidates that is guaranteed to contain a limited number of incorrect answers. Standard conformal prediction provides the ability to adapt to model uncertainty by constructing a calibrated candidate set in place of a single prediction, with guarantees that the set contains the correct answer with high probability. In order to obey this coverage property, however, conformal sets can often become inundated with noisy candidates—which can render them unhelpful in practice. This is particularly relevant to large-scale settings where the cost (monetary or otherwise) of false positives is substantial, such as for in-silico screening for drug discovery, where any positively identified molecular compound is then manufactured and tested. We propose to trade coverage for precision by enforcing that the presence of incorrect candidates in the predicted conformal sets (i.e., the total number of false positives) is bounded according to a user-specified tolerance. Subject to this constraint, our algorithm then optimizes for a generalized notion of set coverage (i.e., the true positive rate) that allows for any number of true answers for a given query (including zero). We demonstrate the effectiveness of this approach across a number of classification tasks in natural language processing, computer vision, and computational chemistry.

## 1   INTRODUCTION

For many classification problems, returning a set of plausible responses instead of a single prediction is a useful way of representing uncertainty (Gammerman and Vovk, 2007; Lei, 2014; Bates et al., 2020). Aligned with this goal, conformal prediction (Vovk et al., 2005) is an increasingly popular method for creating confident prediction sets that provably contain the correct answer with high probability. Unfortunately, these guarantees do not come for free; in order to achieve proper coverage on difficult tasks, conformal predictors are often unable to rule out an overwhelming number of candidates—making their prediction sets large and inefficient (Angelopoulos et al., 2021c). This can make conformal predictors unusable in settings in which the cost of false positives is substantial.

Consider the example of in-silico screening for drug discovery (see Figure 1). In-silico screening uses computational tools to identify potentially viable molecular compounds for a particular purpose of interest. For instance, Stokes et al. (2020) recently performed a high-throughput screen of compounds from the multi-million molecule ZINC15 database (Sterling and Irwin, 2015) using models predictive of *E. coli* inhibition in order to find a new antibiotic, and were successful. Clearly, guaranteeing high hit rates for this task with conformal prediction is an attractive proposition. The catch, however, is that each returned candidate must be verified experimentally—and too many false positives can quickly consume the available budget (e.g., of time, funding, or other limited resources). This is especially relevant when a valid answer—in this case, an effective drug—might not even exist.

In this paper, we are interested in creating confident prediction sets that trade off coverage for precision in order to provably constrain the total *number of false positives* (FP). In other words, we shift the focus of our performance guarantee to be on limiting the number of incorrect answers in our outputs (e.g., to $\leq k$ on average), with the understanding that we can potentially fail to recover some proportion of the true answers—i.e., we may obtain a lower *true positive rate* (TPR). Concretely, suppose we have been given $n$ (possibly) multi-label classification examples $(X_i, Z_i) \in \mathcal{X} \times 2^{\mathcal{Y}}$, $i = 1, \ldots n$ as calibration data, that have been drawn exchangeably from some underlying distribution $P_{XZ}$. Here we consider the generalized scenario where each observation $X_i$ may have any number

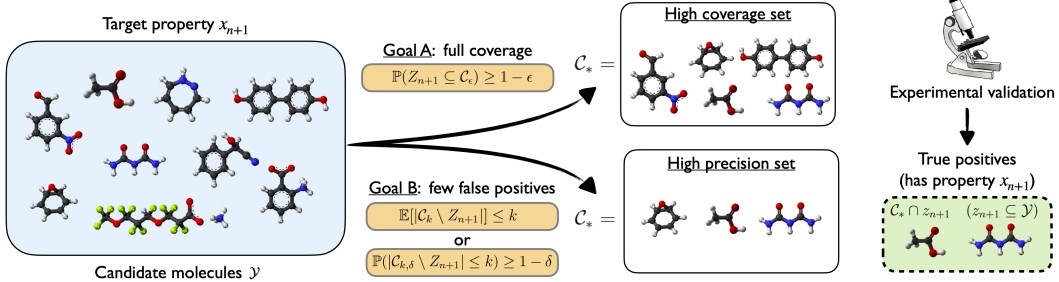

Figure 1: A demonstration of our proposed approach to trading off standard coverage guarantees ("Goal A") for rigorous limits on the total number of false positives included in the output $\mathcal{C}_{k,\delta}$ ("Goal B"). For in-silico screening for drug discovery, limiting false positives is critical when balancing a budget for experimental validation—even if it means that some true positives may be missed.

of corresponding correct labels (including zero, in the case of having no answer at all). That is, the response $Z_i$ can be any subset of the full label space $\mathcal{Y}$.[1] Let $X_{n+1} \in \mathcal{X}$ be a new exchangeable test example for which we would like to predict $Z_{n+1} \subseteq \mathcal{Y}$. Our goal is to construct a set predictor $\mathcal{C}_k(X_{n+1})$ that maximizes recall with respect to the true $Z_{n+1}$, but more importantly has an expected false positive budget that is controlled at a user-defined tolerance level $k \in \mathbb{R}_{>0}$, i.e.,

$$\text{maximize } \mathbb{E}\left[\frac{|\mathcal{C}_k(X_{n+1}) \cap Z_{n+1}|}{\max(|Z_{n+1}|, 1)}\right] \quad \text{s.t.} \quad \mathbb{E}\Big[|\mathcal{C}_k(X_{n+1}) \setminus Z_{n+1}|\Big] \leq k. \tag{1}$$

As an additional alternative to bounding the expected number of false positives to $k$, we also seek a set predictor $\mathcal{C}_{k,\delta}$ that has more direct control of the probability of *exceeding* $k$ false positives:

$$\text{maximize } \mathbb{E}\left[\frac{|\mathcal{C}_{k,\delta}(X_{n+1}) \cap Z_{n+1}|}{\max(|Z_{n+1}|, 1)}\right] \quad \text{s.t.} \quad \mathbb{P}\Big(|\mathcal{C}_{k,\delta}(X_{n+1}) \setminus Z_{n+1}| \leq k\Big) \geq 1 - \delta, \tag{2}$$

where $\delta \in (0, 1)$ is another user-defined tolerance level. Both constructions define different, but useful, operating conditions; the first is more straightforward (e.g., for the general practitioner), while the second offers a finer, two-parameter level of control. Both are well-known concepts in the context of multiple hypothesis testing—i.e., PFER, the per-family error rate, and $k$-FWER, the familywise error rate (Spjøtvoll, 1972; Romano and Wolf, 2007)—though the setting (multi-label classification) and available tools (conformal prediction) considered here differ substantially (see §2). Note that both constraints are marginal over the choice of calibration and test data $\{(X_i, Z_i)\}_{i=1}^{n+1}$, however, enforcing calibration-conditional constraints requires only minor modification (Bates et al., 2020).

In order to achieve the desired levels of false positive control, we present an approach that is based on *set classification*, combined with a form of conformal calibration. Specifically, we use a set nonconformity measure $\mathcal{F} \colon \mathcal{X} \times 2^{\mathcal{Y}} \to \mathbb{R}$ to score candidate output sets, $\mathcal{S} \in 2^{\mathcal{Y}}$. We learn this function from separate multi-label classification training data. Intuitively, a *high* nonconformity score should reflect the confidence that the candidate set might contain a *high* number of false positives, and vice-versa. To empirically maximize the true positive rate, we then return the largest set among all candidates that have nonconformity scores below a threshold that we set such that our false positive constraints are satisfied. Candidate sets are explored greedily with a best-first strategy that adds top-ranked individual labels to a growing, nested output set $\mathcal{S}$ until our nonconformity threshold is met. This both allows us to find efficient solutions for large label spaces $\mathcal{Y}$, and to leverage prior theory for calibrating expectations of monotonic losses for set predictors (Angelopoulos et al., 2021b).

**Contributions.** In summary, our main results are as follows:

- A theoretical reframing of conformal prediction that provides rigorous false positive control;

- A simple and effective strategy for constructing valid output sets with high true positive rates;

- A demonstration of the practical utility of our framework across a range of classification tasks.

---

[1]Note that the classification setting typically considered by conformal prediction, where the correct class is assumed to both exist and be unique, is a special case of this paradigm (i.e., the answer set is always of size one).

## 2 RELATED WORK

**Uncertainty estimation.** In recent years, there has been a growing interest in estimating model uncertainty. A large body of work focuses on calibrating model-based conditional probabilities, $p_\theta(\hat{y}_{n+1}|x_{n+1})$, such that the accuracy, $y_{n+1} = \hat{y}_{n+1}$, is equal to the estimated probability (Niculescu-Mizil and Caruana, 2005; Kuleshov et al., 2018; Kumar et al., 2019). In theory, these estimates could be used to create prediction sets with few false positives, but they are not always guaranteed to be accurate (Guo et al., 2017; Ashukha et al., 2020; Hirschfeld et al., 2020). In a similar vein, Bayesian formalisms underlie several popular approaches to quantifying predictive uncertainty via computing the posterior predictive distribution over model parameters (Neal, 1996; Graves, 2011; Hernández-Lobato and Adams, 2015; Gal and Ghahramani, 2016). However, the quality of these methods can vary depending on the suitability of the presumed prior and on any approximation error.

**Conformal prediction.** As introduced in §1, conformal prediction (Vovk et al., 2005) provides a finite-sample, distribution-free method for obtaining prediction sets $\mathcal{C}$ with guarantees on the event $\mathbf{1}\{Y_{n+1} \in \mathcal{C}(X_{n+1})\}$. Most efforts in CP focus on improving the predictive efficiency, $\mathbb{E}[|\mathcal{C}(X_{n+1})|]$, of the conformal sets (Vovk et al., 2016; Sadinle et al., 2019; Romano et al., 2020; Angelopoulos et al., 2021c; Fisch et al., 2021; Hoff, 2021). As coverage is guaranteed by design, improving efficiency will naturally lead to more precise sets with fewer false positives—but not to a specifiable level. Cauchois et al. (2021) develop a conformal approach to multi-label classification that can guarantee that the prediction set only contains true labels (i.e., FP = 0), but does not offer fine-grained control. Most relevant to our work, Bates et al. (2020) gives a flexible framework for controlling the risk, $\mathbb{E}[\mathcal{L}(Y, \mathcal{T}(X))]$, of a set-valued predictor $\mathcal{T}$ with an arbitrary loss function $\mathcal{L}$—as long the loss respects a monotonic *nesting* property, $\mathcal{S} \subset \mathcal{S}' \Rightarrow \mathcal{L}(\mathcal{S}) \geq \mathcal{L}(\mathcal{S}')$, for any two prediction sets $\mathcal{S}$ and $\mathcal{S}'$. The calibration strategy we use here for marginal expectations is based on the recent extension in Angelopoulos et al. (2021b). Concurrent to this work, Angelopoulos et al. (2021a) proposed methods to rigorously control non-monotonic losses, including the related *false discovery rate* (FDR), which normalizes the number of false positives over the size of the prediction set. However, as most of our target applications have relatively few true positives, FDR control can be somewhat volatile and lead to many empty predictions (making controlling total false positives a more natural fit for this work).

**Multiple testing.** Controlling the number of false positives/discoveries over a collection of hypothesis tests is a well-studied problem in statistics (Dunn, 1961; Benjamini and Hochberg, 1995; Lehmann and Romano, 2005; Romano and Wolf, 2007, *etc.*). Recently, FDR control has also been studied for outlier detection in a conformal inference setting (Bates et al., 2021). Most statistical approaches to false discovery control operate over p-values for each hypothesis test that have specific dependency structures (e.g., independent or positively dependent), or otherwise use more conservative corrections. Though similar, our multi-label classification setting is slightly different from standard multiple testing in that there is (1) an unknown dependency structure between candidate labels for the same query, but also (2) an extra layer of exchangeability over the $n + 1$ queries. Our approach is able to ignore (1) by leveraging (2) within a conformal calibration framework to obtain desirable guarantees.

**Selective classification.** Our work also bears some relation to *selective classification* (El-Yaniv and Wiener, 2010), where models have the option to abstain from answering. In particular, Geifman and El-Yaniv (2017) propose a strategy for finding classifiers with specific selective risks (i.e., the expected accuracy over *answered* examples). In our setting, this is analogous to controlling false positives using $k \approx 0$. If uncertain, the model would have to "abstain" by outputting an empty set.

## 3 BACKGROUND

We begin with a review of conformal prediction (see Shafer and Vovk, 2008). Here, and in the rest of the paper, upper-case letters ($X$) denote random variables; lower-case letters ($x$) denote constants, and script letters ($\mathcal{X}$) denote sets, unless otherwise specified. Proofs are deferred to the appendix.

Given a new example $x$, for every candidate label $y \in \mathcal{Y}$, *single label* conformal prediction (where $y_{n+1}$ is a scalar) either accepts or rejects the null hypothesis that the pairing $(x, y)$ is correct. The test statistic for this test is a *nonconformity measure*, $\mathcal{M}((x, y), \mathcal{D})$, where $\mathcal{D}$ is a dataset of exchangeable, labeled examples. Informally, a lower value of $\mathcal{M}$ reflects that point $(x, y)$ "conforms" to $\mathcal{D}$, whereas a higher value of $\mathcal{M}$ reflects that $(x, y)$ is atypical relative to $\mathcal{D}$. A practical choice for $\mathcal{M}$ is a model-based loss, e.g., $-\log p_\theta(y|x)$, where $\theta$ is a model fit to $\mathcal{D}$. For conformal prediction to work, is important that $\mathcal{M}$ preserves the exchangeability of its inputs: i.e., $\mathcal{M}$ should be symmetric with respect to permutations. In order to avoid retraining $\mathcal{M}$ every time a new candidate label $y$ is

considered, "split" conformal prediction (Papadopoulos, 2008) uses a proper training set $\mathcal{D}_{\text{train}}$ to learn a fixed $\mathcal{M}$. This preserves exchangeability of the calibration and test points without further modification—and is a popular and computationally efficient strategy (which we follow in this work).

To construct a prediction set for a new test point $x_{n+1}$, the conformal classifier outputs all $y$ for which the null hypothesis (that pairing $(x, y)$ is correct) is not rejected. This is achieved by comparing the scores of the test candidates to the scores computed over the first $n$ calibration examples.

**Theorem 3.1** (Split CP, Vovk et al. (2005); Papadopoulos (2008)). *Assume that examples $(X_i, Y_i) \in \mathcal{X} \times \mathcal{Y}$, $i = 1, \ldots, n+1$ are drawn exchangeably from a distribution $P_{XY}$. For a fixed nonconformity measure $\mathcal{M}$, let random variable $V_i = \mathcal{M}(X_i, Y_i)$ be the nonconformity score of $(X_i, Y_i)$. For $\epsilon \in (0, 1)$, define the conformal set (based on the first $n$ examples) at $x \in \mathcal{X}$ as*

$$\mathcal{C}_\epsilon(x) := \left\{ y \in \mathcal{Y} \colon \mathcal{M}(x, y) \leq \text{Quantile}(1 - \epsilon; V_{1:n} \cup \{\infty\}) \right\}. \tag{3}$$

*Then $\mathcal{C}_\epsilon(X_{n+1})$ satisfies $\mathbb{P}(Y_{n+1} \in \mathcal{C}_\epsilon(X_{n+1})) \geq 1 - \epsilon$.*

**Remark 3.2.** Cauchois et al. (2021) recently extended conformal prediction to the multi-label case to obtain guarantees of the form $\mathbb{P}(\mathcal{C}_\epsilon^{\text{inner}}(X_{n+1}) \subseteq Z_{n+1} \subseteq \mathcal{C}_\epsilon^{\text{outer}}(X_{n+1})) \geq 1 - \epsilon$. Their approach constructs bounding sets using a modified version of the single label algorithm (i.e., Eq. (3)).

The motivation for our work is quickly evident from Eq. (3): if we are unable to reject most candidates $y \in \mathcal{Y}$ based on their nonconformity scores $\mathcal{M}(x_{n+1}, y)$, then $\mathcal{C}_\epsilon(x)$ can contain many false positives.

## 4 CONFORMAL PREDICTION WITH FALSE POSITIVE CONTROL

We now introduce our strategy for trading off *coverage* for *precision* by imposing constraints on the number of false positives that are contained in our output sets. To briefly remind the reader of our setting, we assume that we have been given $n$ exchangeable multi-label classification examples, $(X_i, Z_i) \in \mathcal{X} \times 2^{\mathcal{Y}}$, $i = 1, \ldots n$ as calibration data, that are drawn from a distribution $P_{XZ}$. The response $Z_i$ is treated as a generalized set of correct labels for input $X_i$, and is a subset of $\mathcal{Y}$, where $\mathcal{Y}$ is finite. For a prediction $\mathcal{C}(x_{n+1}) \subseteq \mathcal{Y}$ evaluated at a point $x_{n+1} \in \mathcal{X}$ with true label set $z_{n+1} \subseteq \mathcal{Y}$, we define the *true positive proportion* (TPP) as the ratio of correct labels that are recovered:

$$\text{TPP}(z_{n+1}, \mathcal{C}(x_{n+1})) := \frac{|\mathcal{C}(x_{n+1}) \cap z_{n+1}|}{\max(|z_{n+1}|, 1)} \qquad \text{(Note that } \text{TPR} := \mathbb{E}[\text{TPP}]\text{),} \tag{4}$$

and the number of *false positives* (FP) as the total count of incorrect labels in $\mathcal{C}(x_{n+1})$:

$$\text{FP}(z_{n+1}, \mathcal{C}(x_{n+1})) := |\mathcal{C}(x_{n+1}) \setminus z_{n+1}|. \tag{5}$$

Our goal, as stated in §1, is to maximize the expected TPP (in standard classification, this would equate to accuracy; or power in statistics), while constraining the FP in either of two ways:

**Definition 4.1** (*k*-FP validity). *A conformal classifier $\mathcal{C}_k$ evaluated at test point $(X_{n+1}, Z_{n+1})$ is $k$-FP valid if the expectation of its FP satisfies $\mathbb{E}[\text{FP}(Z_{n+1}, \mathcal{C}_k(X_{n+1}))] \leq k$.*

**Definition 4.2** $((k, \delta)$-FP validity). *A conformal classifier $\mathcal{C}_{k,\delta}$ evaluated at test point $(X_{n+1}, Z_{n+1})$ is $(k, \delta)$-FP valid if the CDF of its FP satisfies $\mathbb{P}(\text{FP}(Z_{n+1}, \mathcal{C}_{k,\delta}(X_{n+1})) \leq k) \geq 1 - \delta$.*

### 4.1 AN ORACLE SET PREDICTOR

To motivate our approach, imagine an *oracle* with access to $P_{Z|X}$, the conditional distribution of the multi-label set $Z$ given the input $X$. Given this information, for any input $x \in \mathcal{X}$ and candidate set $\mathcal{S} \in 2^{\mathcal{Y}}$, such an oracle would be able to exactly calculate both the expectation and the conditional distribution of the number of false (and true) positives in $\mathcal{S}$ given $x$. In order to maximize the TPR while meeting $k$-FP and $(k, \delta)$-FP validity, it could then yield the following set predictions:

$$\mathcal{C}_k^{\text{oracle}}(x) := \underset{\mathcal{S} \subseteq 2^{\mathcal{Y}}}{\arg\max} \left\{ \mathbb{E}[\text{TPP}(\mathcal{S}) \mid x] \colon \mathbb{E}[\text{FP}(Z, \mathcal{S}) \mid x] \leq k \right\}, \quad \text{and} \tag{6}$$

$$\mathcal{C}_{k,\delta}^{\text{oracle}}(x) := \underset{\mathcal{S} \subseteq 2^{\mathcal{Y}}}{\arg\max} \left\{ \mathbb{E}[\text{TPP}(\mathcal{S}) \mid x] \colon \mathbb{P}(\text{FP}(Z, \mathcal{S}) \mid x) > k) < \delta \right\}, \tag{7}$$

where ties in the $\arg\max$ are broken by set size (smaller is better). This oracle has the advantage of not only being marginally valid, but also being *conditionally* valid given the test input $X_{n+1} = x_{n+1}$.

---

**Algorithm 1** Conformal prediction with false positive control (in expectation case, see Eq. (1)).

---

**Definitions:** $x_{n+1}$ is the test point, $\mathcal{D}_{\text{train}}$ is a training set, $\mathcal{D}_{\text{cal}}$ is an exchangeable calibration set, $k$ is the FP tolerance, and $B$ is a parameter for considering only the top individually ranked candidates $y \in \mathcal{Y}$. MultilabelModel is an abstract model that provides individual label predictions. SetModel is an abstract FP-prediction model operating over sets (where in this work we propose to use DeepSets (Zaheer et al., 2017)).

1: **function** TRAIN($\mathcal{D}_{\text{train}}$, $k$)
2:    $\mathcal{D}_{\text{train}}^{(1)}, \mathcal{D}_{\text{train}}^{(2)} \leftarrow$ SPLIT($\mathcal{D}_{\text{train}}$)
3:    *# Use part of the training set to learn individual label likelihood $p_\theta(y_c \in Z \mid x)$.*
4:    $p_\theta(y_c \in Z \mid x) \leftarrow$ TRAIN(MultilabelModel, $\mathcal{D}_{\text{train}}^{(1)}$)
5:    *# Use the other (smaller) part to learn the FP-predictive set function $\mathcal{F}(x, \mathcal{S})$.*
6:    $\mathcal{F}(x, \mathcal{S}) \leftarrow$ TRAIN(SetModel, $\mathcal{D}_{\text{train}}^{(2)}$)
7:    *# Manually define $\mathcal{F}$ to be monotonic for nested sets $\mathcal{S}_1 \subseteq \ldots \subseteq \mathcal{S}_m$.*
8:    $\widetilde{\mathcal{F}}(x, \mathcal{S}_m) \leftarrow \max\{\mathcal{F}(x, \mathcal{S}_1), \ldots, \mathcal{F}(x, \mathcal{S}_m)\}$
9:    **return** $p_\theta, \widetilde{\mathcal{F}}$
10: **function** CALIBRATE($p_\theta, \widetilde{\mathcal{F}}, \mathcal{D}_{\text{cal}}, k, B$)
11:    $\mathcal{T}_{\text{cal}} = \{\}$
12:    **for** $(x_i, z_i) \in \mathcal{D}_{\text{cal}}$ **do**
13:       *# Rank candidates $y \in \mathcal{Y}$ by top $p_\theta(y_c \in Z_i \mid x_i)$, up to bound $B$.*
14:       $\{\mathcal{S}_{i,1}, \ldots, \mathcal{S}_{i,B}\} \leftarrow \{$SORT($\mathcal{Y}, p_\theta(y_c \in Z_i \mid x_i))_{1:j} : j \in \{1, \ldots, B\}\}$
15:       *# Implicitly define MFP based on this ordering.*
16:       MFP($x_i, z_i$) $\leftarrow$ INIT($x_i, z_i, \widetilde{\mathcal{F}}, \{\mathcal{S}_1, \ldots, \mathcal{S}_B\}$)
17:       $\mathcal{T}_{\text{cal}} \leftarrow \mathcal{T}_{\text{cal}} \cup \{$MFP($x_i, z_i$)$\}$
18:    $t_k \leftarrow$ FIND_THRESHOLD($\mathcal{T}_{\text{cal}}, B, k$)   *# Using Eq. (10).*
19:    **return** $t_k$
20: **function** PREDICT($x_{n+1}, p_\theta, \widetilde{\mathcal{F}}, t_k, B$)
21:    *# Rank new candidates $y \in \mathcal{Y}$ by top $p_\theta(y_c \in Z_{n+1} \mid x_{n+1})$, up to bound $B$ (as in line 14).*
22:    $\{\mathcal{S}_{n+1,1}, \ldots, \mathcal{S}_{n+1,B}\} \leftarrow \{$SORT($\mathcal{Y}, p_\theta(y_c \in Z_{n+1} \mid x_{n+1}))_{1:j} : j \in \{1, \ldots, B\}\}$
23:    *# Return largest prediction set that passes threshold $t_k$.*
24:    $\mathcal{C}_k(x_{n+1}) \leftarrow \arg\max_{\mathcal{S} \in \{\mathcal{S}_{n+1,1} \ldots, \mathcal{S}_{n+1,B}\}} \{|\mathcal{S}| : \widetilde{\mathcal{F}}(x_{n+1}, \mathcal{S}) \leq t_k\}$
25:    **return** $\mathcal{C}_k(x_{n+1})$

---

## 4.2 APPROXIMATING THE ORACLE USING SET FUNCTIONS

Of course, computing such an oracle is not practical, as $P_{Z|X}$ is not known. Instead, let $\mathcal{F}$ be a set function $\mathcal{F} : \mathcal{X} \times 2^{\mathcal{Y}} \to \mathbb{R}$ that generates a score for the candidate set $\mathcal{S}$ given $x_{n+1}$. Importantly, $\mathcal{F}$ can be any function (as we will later show), but to best replicate the oracle's behavior for $k$-FP and $(k, \delta)$-FP valid predictions, $\mathcal{F}$ should be a good estimator of the distribution of false positives in $\mathcal{S}$.

We now describe a simple approach to modeling $\mathcal{F}$ using DeepSets (Zaheer et al., 2017). DeepSets is a popular method which is known to be a universal approximator for continuous set functions. Let $\{\phi(x, y_1), \ldots, \phi(x, y_s)\}$ featurize a candidate set $\mathcal{S} \subseteq \mathcal{Y}$, where $\phi(x, y_c) \in \mathbb{R}^d$ is a function of $(x, y_c)$, for $y_c \in \mathcal{S}$. In practice, we find that taking $\phi(x, y_c)$ to be a (fixed) estimate of $p_\theta(y_c \in Z \mid x)$, the marginal likelihood of $y_c$ being a correct label, performs well and is simple to implement. These (uncalibrated) prediction scores can be provided by any backbone model. For example, in our in-silico screening task, we define $\phi$ using a directed MPNN (Yang et al., 2019) that independently classifies molecules as active or inactive for the target property. Given $\phi$, the DeepSets model is defined as

$$\Psi(x, \mathcal{S}) := \text{softmax}\Big(\text{dec}\Big(\sum_{y_c \in \mathcal{S}} \text{enc}(\phi(x, y_c); \theta_1)); \theta_2\Big)\Big), \tag{8}$$

where $\text{enc}(\cdot)$ and $\text{dec}(\cdot)$ are two neural encoder/decoder models parameterized by $\theta_1$ and $\theta_2$, respectively, and $\text{softmax}(\cdot)$ is taken over the total possible number of false positives, $\{0, \ldots, |\mathcal{Y}|\}$. $\Psi$ is trained to predict the number of false positives in $\mathcal{S}$ via cross entropy, using labeled sets sampled from held-out training data. We then compute $\mathcal{F}_k$ and $\mathcal{F}_{k,\delta}$ (for either $k$-FP or $(k, \delta)$-FP validity) as

$$\mathcal{F}_k(x, \mathcal{S}) := \sum_{\eta=0}^{|\mathcal{S}|} \eta \cdot \Psi(x, \mathcal{S})_\eta \quad \text{and} \quad \mathcal{F}_{k,\delta}(x, \mathcal{S}) := 1 - \sum_{\eta=0}^{\min(k, |\mathcal{S}|)} \Psi(x, \mathcal{S})_\eta, \tag{9}$$

where $\Psi(x, \mathcal{S})_\eta$ denotes the $\eta$-th index of the $\text{softmax}$ (i.e., the estimated probability that FP $= \eta$). In the next section we will refer only to $\mathcal{F}$, with the understanding it differs based on the validity goal.

### 4.3 Identifying valid candidate sets

Although our set predictor $\mathcal{F}$ is trained to model either the expected FP or its CDF, it is not necessarily accurate. If $\mathcal{F}$ were simply substituted into Eq. (6) or Eq. (7), it may not produce valid set predictions. To remedy this, we carefully calibrate a threshold for accepting candidate sets based on $\mathcal{F}$. First, we greedily identify a sequence of *nested* candidate sets, $\mathcal{S}_1 \subset \mathcal{S}_2 \subset \ldots \subset \mathcal{Y}$, by ranking individual labels $y_c \in \mathcal{Y}$ by their estimated likelihoods of being true positives, i.e., $p_\theta(y_c \in Z \mid x)$, and include them one by one. Next, each of these candidate sets is given a score by $\mathcal{F}$, which we treat as a "nonconformity score." We manually enforce that the scores produce a ranking consistent with their size: for set $\mathcal{S}_j$ we compute its adjusted nonconformity score $\widetilde{\mathcal{F}}(x, \mathcal{S}_j)$ as $\widetilde{\mathcal{F}}(x, \mathcal{S}_j) := \max\{\mathcal{F}(x, \mathcal{S}_1), \ldots, \mathcal{F}(x, \mathcal{S}_j)\}$. For efficiency (see Remark 4.4), we only take the top $B \leq |\mathcal{Y}|$ sets (although this doesn't come for free: a smaller $B$ may result in fewer true positives).

Let $\mathrm{MFP}(x, z, t) := \max\big\{\mathrm{FP}(z, \mathcal{S}_j)\colon \widetilde{\mathcal{F}}(x, \mathcal{S}_j) \leq t\big\}$ denote the maximum number of false positives over all nested sets $\mathcal{S}_j$ for an input $x$ that have nonconformity scores less than $t$. If this thresholded set is empty, then the MFP is defined to be $0$. Since the sets are ordered by size, this quantity is guaranteed to be monotonically non-decreasing in $t$, i.e., $t \leq t' \implies \mathrm{MFP}(x, z, t) \leq \mathrm{MFP}(x, z, t')$. Using this property, we can calibrate $t$ to create a set filter with controlled MFP, as formalized next:

**Theorem 4.3** (FP-CP). *Assume that examples $(X_i, Z_i) \in \mathcal{X} \times 2^{\mathcal{Y}}$, $i = 1, \ldots, n + 1$ are drawn exchangeably from a distribution $P_{XY}$. For each example $i$, let $S_{i,j}$, $j = 1, \ldots, B$ (where $B \leq |\mathcal{Y}|$ is a finite hyper-parameter) be nested sets with non-decreasing, lower-bounded nonconformity scores $\widetilde{\mathcal{F}}(X_i, \mathcal{S}_{i,j})$. For $k \in \mathbb{R}_{>0}$ and $\delta \in (0, 1)$ define the random variables $T_k$ and $T_{k,\delta}$ as*

$$T_k := \sup\left\{t \in \mathbb{R}\colon \frac{1}{n+1}\left(B + \sum_{i=1}^n \mathrm{MFP}(X_i, Z_i, t)\right) \leq k\right\} \quad and \tag{10}$$

$$T_{k,\delta} := \sup\left\{t \in \mathbb{R}\colon \frac{1}{n+1}\sum_{i=1}^n \mathbf{1}\{\mathrm{MFP}(X_i, Z_i, t) \leq k\} \geq 1 - \delta\right\}. \tag{11}$$

*Then for any $j \in \{1, \ldots, B\}$, we have that $\mathbb{E}[\mathrm{FP}(Z_{n+1}, \mathcal{S}_{n+1,j}) \mid \widetilde{\mathcal{F}}(X_i, \mathcal{S}_{n+1,j}) \leq T_k] \leq k$, and $\mathbb{P}(\mathrm{FP}(Z_{n+1}, \mathcal{S}_{n+1,j}) \leq k \mid \widetilde{\mathcal{F}}(X_i, \mathcal{S}_{n+1,j}) \leq T_{k,\delta}) \geq 1 - \delta$.*

The proof, given in Appendix A, uses results from marginal RCPS (Angelopoulos et al., 2021b).

**Remark 4.4.** The hyper-parameter $B$ plays an important role. $T_k$ may be very conservative if $B = |\mathcal{Y}|$ and $|\mathcal{Y}|$ is very large, to the point where $T_k = -\infty$ always if $|\mathcal{Y}| > k * (n + 1)$. It can be beneficial to truncate the considered label space $\mathcal{Y}$ for an example $x_i$ to only the top $B \ll k * (n+1)$ individual candidates, $\{y_1, \ldots, y_B\} \in \mathcal{Y}^B$. For example, for text generation tasks (like machine translation), the label-space $\mathcal{Y}$ is infinite, but we can instead restrict our predictions to a subset of the top $B$ beam search candidates (where $B$ can still be large to a reasonable degree).

**Remark 4.5.** Note that no constraints are put on the underlying $\widetilde{\mathcal{F}}$ in Theorem 4.3; it is model-agnostic, and need not even be a DeepSets architecture of the form of Eq. (8). If, however, $\widetilde{\mathcal{F}}$ is a good estimator of $\mathrm{FP}(Z, \mathcal{S}) \mid X$, then $T_k$ and $T_{k,\delta}$ are likely to identify sets that are approximately valid conditional on $X_{n+1}$, which we show empirically for our choice of DeepSets models in §6.

### 4.4 Selecting the final output set

The main result of Theorem 4.3 is that, using the calibrated nonconformity threshold $T_k$ or $T_{k,\delta} = t^*$, we can construct a collection of sets that are simultaneously valid. In other words, we are free to select *any* set in the filtered set of candidates $\{\mathcal{S}_{n+1,j}\colon \widetilde{\mathcal{F}}(x, \mathcal{S}_{n+1,j}) \leq t^*\}$ as a valid output. As a greedy, but simple and robust, approach we simply take the *largest* set in our filtered set of candidates as our final output. That is, for an input $x \in \mathcal{X}$ and calibrated threshold $t^* \in \mathbb{R}$

$$\mathcal{C}_\circ(x, t^*) := \underset{\mathcal{S} \in \{\mathcal{S}_1 \ldots, \mathcal{S}_B\}}{\arg\max} \left\{|\mathcal{S}|\colon \widetilde{\mathcal{F}}(x, \mathcal{S}) \leq t^*\right\} \tag{12}$$

where $\mathcal{C}_\circ$ is either $\mathcal{C}_k$ or $\mathcal{C}_{k,\delta}$ (i.e., they are constructed the same way, but for different thresholds $t^*$), and where we define $\arg\max$ to return $\varnothing$ if the set $\{\mathcal{S} \in \{\mathcal{S}_1, \ldots, \mathcal{S}_B\}\colon \widetilde{\mathcal{F}}(x, \mathcal{S}) \leq t^*\}$ is empty.

**Proposition 4.6** (Greedy FP-CP). *As defined in Eq. (12), the set $\mathcal{C}_k(X_{n+1}, T_k)$ is $k$-FP valid, and the set $\mathcal{C}_{k,\delta}(X_{n+1}, T_{k,\delta})$ is $(k, \delta)$-FP valid. Furthermore, among sets $\mathcal{S}_{n+1,j} \in \{\mathcal{S}_{n+1,1}, \ldots, \mathcal{S}_{n+1,|\mathcal{Y}|}\}$ that have nonconformity scores $\leq T_\circ = T_k$ or $T_{k,\delta}$, the set $\mathcal{C}_\circ(X_{n+1}, T_\circ)$ maximizes the TPR.*

## 5 EXPERIMENTAL SETUP

In this section, we briefly outline our evaluation tasks and associated models. We also describe our evaluation and baselines. For all experiments, we set $B = 100$ (recall Remark 4.4), which caps the number of considered labels per example to the top 100. Appendix C contains additional details.

### 5.1 TASKS

**In-silico screening for drug discovery.** As introduced in §1, the goal of in-silico screening is to identify potentially effective drugs to manufacture and test. We use the ChEMBL database (Mayr et al., 2018) to screen molecules for combinatorial constraint satisfaction, where given a constraint such as "*has property A but not property B*," we want to identify the subset of molecules from a given set of candidates that have the desired attributes. We partition the dataset both by molecules and property combinations, so that at test time the model must make predictions on combinations it has never been tested on before, over a pool of molecules that it has never seen before. Scores for individual candidate molecules are obtained via combining independent property assignment probabilities predicted by an ensemble of directed MPNNs (Yang et al., 2019) using a naive independence assumption.

**Object detection.** The goal of object detection is to place bounding boxes around all objects of a certain type that are present in an image (of which there may be few, many, or none). We use the MS-COCO dataset (Lin et al., 2014), a large scale object detection dataset with images of complex everyday scenes containing 80 object categories (such as person, bicycle, dog, car, etc). We extract typed bounding box candidates (i.e., tuples of both location *and* category) using an EfficientDet model (Tan et al., 2020) with non-maximum suppression. True positives are defined as boxes that have an intersection over union (IoU) $> 0.5$ with a gold annotation of the same type.

**Entity extraction.** In entity extraction, we are interested in identifying all named entities that appear in a tokenized sentence $x$ of length $l$, $x = \{w_1, \ldots, w_l\}$. A named entity is a contiguous span $\{w_{\text{start}}, \ldots, w_{\text{end}}\} \subseteq x$ of the input sentence that refers to a real-world object (such as a person, location, organization, or product) that can be denoted with a proper name. We report results on the CoNLL NER dataset (Tjong Kim Sang and De Meulder, 2003), where we use the PURE span-based entity extraction model of Zhong and Chen (2021) to predict scores for all $\mathcal{O}(l^2)$ candidate spans. We consider exact span predictions of the correct category to be true positives, and all others to be false positives. Not all sentences are guaranteed to have entities (in fact, many do not contain any).

### 5.2 EVALUATION

We use a proper training, validation, and test set for each task. We use the training set to learn all models, i.e., $p_\theta(y_c \in Z \mid x)$ and $\mathcal{F}$.[2] We do model selection on the validation set, and report final numbers as the average over 1000 random trials on the test set, where in each trial we partition the data into 80% calibration ($x_{1:n}$) and 20% prediction points ($x_{n+1}$). To compare aggregate performance across tolerance levels, we plot each metric as a function of $k$ (up to $k = B$), and compute the area under the curve (AUC). Shaded regions show the 16-84th percentiles across trials. In addition to TPR and FP, inspired by Angelopoulos et al. (2021c), we compute the *size-stratified $k$-FP violation*:

$$\text{SSFP}_k(\mathcal{C}, \{\mathcal{A}\}_{s=1}^a) := \sup_s \max \left( \widehat{\mathbb{E}}[\text{FP}(Z_{n+1}, \mathcal{C}_k(X_{n+1})) \mid |\mathcal{C}_k(X_{n+1})| \in \mathcal{A}_s] - k, 0 \right) \quad (13)$$

where $\{\mathcal{A}\}_{s=1}^a$ forms a partition of $\{1, \ldots, |\mathcal{Y}|\}$, and $\widehat{\mathbb{E}}$ denotes the empirical average over our test trials. $\text{SSFP}_{k,\delta}$ for $(k, \delta)$-FP violation is defined similarly, where we compare the worst-case average number of predictions with more than $k$ false positives to the tolerance $\delta$. As argued in Angelopoulos et al. (2021c), a lower size-stratified violation suggests that a classifier has better conditional coverage. As marginal validity is already theoretically guaranteed by our procedure, we focus on this metric.

### 5.3 BASELINES

For all experiments, we compare our conformalized DeepSets model (NN), to the following baselines:

1. **Top-k.** We naively take the top $k'$ fixed predictions for any $x_{n+1}$, where $k'$ is found using average performance on the calibration set (without any correction factors, so it is *not* generally guaranteed to be valid). Note that $k'$ can be (and mostly is) different than the user-specified $k$ for FP.

---

[2]If training data is limited, we learn the DeepSets model for $\mathcal{F}$ with fixed hyper-parameters on the validation data, after we have finished performing any other model selection for the base multi-label model, $p_\theta(y_c \in Z \mid x)$.

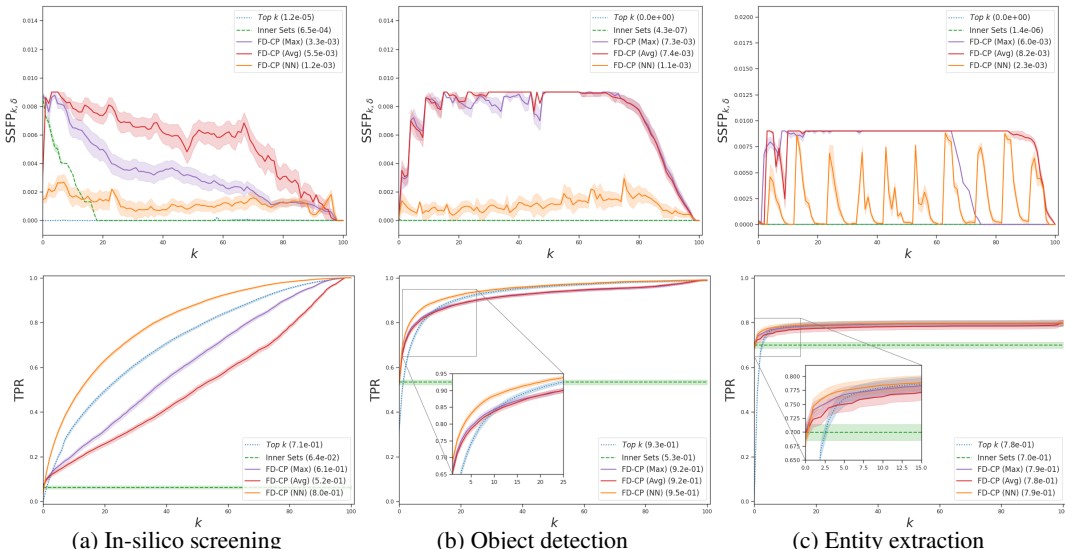

Figure 2: $(k, \delta)$-FP results as a function of $k$ for $\delta = 0.1$ (up to $k = B = 100$). The top row plots $\text{SSFP}_{k,\delta}$ violation (lower is better). The bottom row plots TPR (higher is better). We see that compared to the other baselines, our conformal deep sets approach (NN) has the best (or close to the best) TPR AUC across tasks, while having the lowest (or close to the lowest) $\text{SSFP}_{k,\delta}$ violation.

2. **Outer Sets @ 90.** We use the (one-sided) multi-label conformal prediction technique of Cauchois et al. (2021) to bound $\mathbb{P}(Z_{n+1} \subseteq \mathcal{C}_\epsilon(X_{n+1})) \geq 0.90$. Though not directly comparable, we use this to benchmark our method against sets that preserve marginal coverage (at a typical level). For simplicity, we use the direct inner/outer method without dynamic CQC quantiles.[3]

3. **Inner Sets.** Again, we use the (one-sided) method of Cauchois et al. (2021), this time to bound $\mathbb{P}(\mathcal{C}_\epsilon(X_{n+1}) \subseteq Z_{n+1}) \geq 1 - \epsilon$ at level $\epsilon = k/B$ (recall that $B \leq |\mathcal{Y}|$ is the truncation parameter, and the FP upper bound) for $k$-FP control and at level $\epsilon = \delta$ for $(k, \delta)$-FP control. It is straightforward to show that these levels of $\epsilon$ conservatively achieve $k$-FP and $(k, \delta)$-FP control.

4. **Independent scoring (max).** We take $\mathcal{F}(x, \mathcal{S})$ to be the maximum individual label uncertainty in $\mathcal{S}$, $\max\{1 - p_\theta(y_c \in Z \mid x) : y_c \in \mathcal{S}\}$. This is equivalent to choosing labels independently. The full model is calibrated using the same FP-CP algorithm (it is a drop-in replacement for the NN).

5. **Average scoring (avg).** We take $\mathcal{F}(x, \mathcal{S})$ to be the average individual label uncertainty in $\mathcal{S}$, $|\mathcal{S}|^{-1} \sum_{y_c \in \mathcal{S}} 1 - p_\theta(y_c \in Z \mid x)$. We calibrate $p_\theta(y_c \in Z \mid z)$ using Platt scaling (Platt, 1999). As with the max score, the full model is calibrated using the same FP-CP algorithm.

Baseline (1) contrasts our approach with what is normally a "first thought" in practice; (2) and (3) test the efficacy of our system over existing techniques; (4) and (5) compare to simpler scoring variants.

## 6 EXPERIMENTAL RESULTS

**Constraining false positives.** The top rows of Figures 2 and 3 show the size-stratified violation (Eq. (13)) for $(k, \delta)$-FP and $k$-FP, respectively. Across levels of $k$, our approach based on DeepSets (NN) achieves substantially lower worst-case violations than either of the max or average score baselines. The Top-$k$ and Inner Sets approaches also prevent large violations (though, by itself, this result is not necessarily impressive—as simply always returning an empty set will lead to zero SSFP). When accounting for TPR (bottom rows), we see that our FP-CP model does considerably better.

**Maximizing true positive rates.** The top rows of Figures 2 and 3 plot TPR rates and AUC across many values of $k$, while Table 1 details results for the in-silico screening task for several representative configurations. On the screening task, we see that our FP-CP (NN) method provides significantly higher TPR than other baselines. For example, allowing no more than 5 false positives leads to a TPR of 36.1% with $k$-FP. In comparison, the TPR of Top k is only 29.8%. Differences in TPR are less pronounced on the object detection and entity extraction tasks, however, all FP-CP methods (with max, avg, or NN scoring) provide high TPR (exceeding non FP-CP methods) even at low values of $k$.

---

[3]Preliminary studies indicated that including CQC quantiles did not lead to significantly different results.

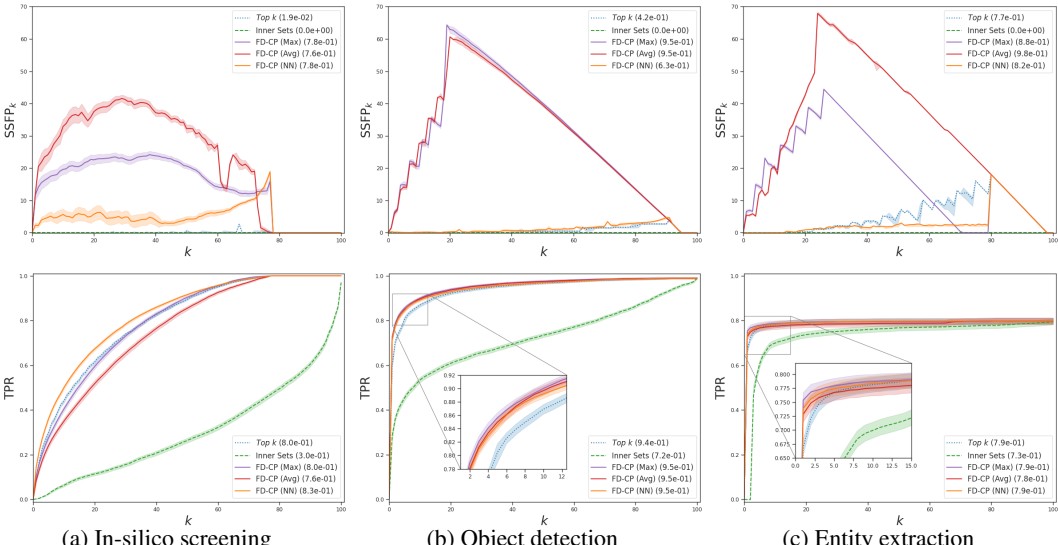

Figure 3: $k$-FP results as a function of $k$ for (up to $k = B = 100$). The top row plots size-stratified $k$-FP violation (lower is better). The bottom row plots the TPR (higher is better). As the number of allowed false positives grows, our methods quickly achieve high power. Consistent with Figure 2, our conformal DeepSets approach (NN) demonstrates high TPR and low $\mathrm{SSFP}_k$ across tasks.

| *k-FP*: | Top k | | Inner Sets | | FP-CP (Max) | | FP-CP (Avg) | | FP-CP (NN) | |
|---|---|---|---|---|---|---|---|---|---|---|
| | *Avg.* FP | *TPR* | *Avg.* FP | *TPR* | *Avg.* FP | *TPR* | *Avg.* FP | *TPR* | *Avg.* FP | *TPR* |
| $k = 5$ | 4.59 | 29.8 | 0.14 | 2.5 | 4.98 | 27.5 | 4.99 | 24.1 | 4.98 | **36.1** |
| $k = 15$ | 14.47 | 53.4 | 0.88 | 9.5 | 14.98 | 50.7 | 14.98 | 43.8 | 14.99 | **59.9** |
| $k = 25$ | 24.51 | 68.0 | 1.49 | 13.4 | 24.98 | 66.8 | 24.98 | 58.9 | 24.99 | **73.2** |
| $k = 35$ | 34.54 | 78.2 | 2.45 | 18.4 | 34.97 | 78.4 | 34.96 | 71.2 | 34.99 | **82.5** |
| $(k, \delta)$-*FP with* $1 - \delta = 0.9$: | | | | | | | | | | |
| | FP $\le k$ | *TPR* | FP $\le k$ | *TPR* | FP $\le k$ | *TPR* | FP $\le k$ | *TPR* | FP $\le k$ | *TPR* |
| $k = 5$ | 100.0 | 20.5 | 96.6 | 6.36 | 90.0 | 15.8 | 90.0 | 14.0 | 90.0 | **31.6** |
| $k = 15$ | 94.7 | 42.4 | 99.5 | 6.36 | 90.0 | 26.7 | 90.0 | 22.3 | 90.0 | **55.3** |
| $k = 25$ | 96.6 | 55.7 | 100.0 | 6.36 | 90.0 | 37.4 | 90.0 | 29.3 | 90.0 | **69.0** |
| $k = 35$ | 97.5 | 66.2 | 100.0 | 6.36 | 90.0 | 49.1 | 90.0 | 36.9 | 90.0 | **79.0** |

Table 1: Results for the in-silico screening task over the ChEMBL dataset. TPR and FP $\le k$ are expressed as percents (i.e., $\times 100$). We see that our FP-CP methods meet our target thresholds; using the Inner Sets formulation does too, but is conservative (as expected). Applying FP-CP calibration with our DeepSets model (see NN) yields substantially higher TPR across various tolerance levels $k$.

**Comparison to conformal coverage methods.** Table B.1 gives the results of the coverage-seeking Outer Sets method at level $0.90$ (a typical tolerance). Indeed, we achieve strong TPR ($97.2\%$ for the in-silico screening task), but also incur a high false positive cost in the process ($63.6$ average FP for in-silico screening). In contrast, our method allows us to directly modulate false positives, without losing high TPR (e.g., equivalently controlling for $\le 63.6$ FP, we acheive $97.0\%$ TPR on screening).

## 7 CONCLUSION

Conformal prediction, in its standard formulation, already grants theoretical performance guarantees that can be critical in many machine learning applications. Naively applying CP, however, can yield disappointing results. Even if the target coverage is indeed upheld, the predicted sets may be too large, and include too many false positives to be practical. In this paper, we proposed a method for trading coverage guarantees in favor of precision guarantees, where we enforce a limit to the number of false positives that are contained in our prediction sets. Our results show that our method yields classifiers that (1) still achieve strong true positive rates compared to their coverage-seeking counterparts, and (2) provide meaningful output sets with effectively controlled total false positive counts.

## ETHICS STATEMENT

Our FP control methods are general and can be applied to many applications and on top of any model for computing nonconformity scores. It's worth noting that any undesirable biases exhibited by underlying models can still propagate to the prediction sets of our methods. While our methods provide marginal performance guarantees, we recommend that any application to perform controlled evaluation across target populations to ensure fairness.

## REPRODUCIBILITY STATEMENT

All datasets used in this paper are publicly available (see §5.1). Also, we use publicly available models for computing the nonconformity scores. For drug discovery, we use the ensemble of directed MPNNs from chemprop (https://github.com/chemprop/chemprop). For object detection, we use tf_efficientdet_d2 from https://github.com/rwightman/efficientdet-pytorch. For entity extraction, we train the PURE entity model (https://github.com/princeton-nlp/PURE) on the ConLL03 dataset using a context window of 100, batch size 32, albert-base-v2 encoding model, and the suggested learning rate of $1e-5$, and task learning rate $5e-4$. Default values were used for all other parameters. The results in Section 6 are all based on the experimental setting described in Section 5. We will release our code for running these experiments and for reproducing all plots and tables.

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

# Appendix

## A  PROOFS

### A.1  PROOF OF THEOREM 3.1

*Proof.* This is a well-known result; we prove it here for completeness. Since the nonconformity scores $V_i$ are constructed symmetrically, then

$$((X_1, Y_1), \ldots, (X_{n+1}, Y_{n+1})) \stackrel{d}{=} ((X_{\sigma(1)}, Y_{\sigma(1)}), \ldots, (X_{\sigma(n+1)}, Y_{\sigma(n+1)}))$$

$$\iff (V_1, \ldots, V_{n+1}) \stackrel{d}{=} (V_{\sigma(1)}, \ldots, V_{\sigma(n+1)})$$

for all permutations $(\sigma(1), \ldots \sigma(n+1))$. Therefore, if $\{(X_i, Y_i)\}_{i=1}^{n+1}$ are exchangeable, then so too are their nonconformal scores $\{V_i\}_{i=1}^{n+1}$.

By the construction of $\mathcal{C}$, we have

$$Y_{n+1} \in \mathcal{C}_k(X_{n+1}) \iff V_{n+1} \leq \text{Quantile}(1 - \epsilon, V_{1:n} \cup \{\infty\}).$$

This implies that $V_{n+1}$ is ranked among the $\lceil (1 - \epsilon) \cdot (n+1) \rceil$ smallest of $V_1, \ldots, V_n, \infty$. Since $V_i$ are exchangeable, this happens with probability at least $1 - \epsilon$. $\square$

### A.2  PROOF OF THEOREM 4.3

Our proof will use Theorem 1 of Angelopoulos et al. (2021a), which we state here:

**Theorem A.1** (Marginal RCPS, monotonic non-increasing case). *Let $L_i \colon \mathbb{R} \to \mathbb{R}$, $i = 1, \ldots, n+1$ be exchangeable functions, where $L_i(t)$ is non-increasing in $t$. Also, take $g \colon \mathbb{R} \to \mathbb{R}$ where $g(x)$ is non-decreasing in $x$. Further assume that $g \circ L_i$ is right-continuous, and*

$$\inf_t g(L_i(t)) < \gamma, \quad \sup_t g(L_i(t)) \leq B < \infty \text{ almost surely.} \tag{14}$$

*For any $\gamma \geq 0$, define the random variable $T(\gamma, g)$ as*

$$T(\gamma; g) := \inf \left\{ t \colon \frac{1}{n+1} \sum_{i=1}^{n} g(L_i(t)) \leq \gamma \right\}. \tag{15}$$

*Then $\mathbb{E}[g \circ L_{n+1}(T(\gamma; g))] \leq \gamma + \frac{B}{n+1}$.*

**Corollary A.2** (Marginal RCPS, adjusted). *Under the same setting as in Theorem A.1,*

$$\mathbb{E}[g \circ L_{n+1}(\widetilde{T}(\gamma; g))] \leq \gamma, \tag{16}$$

*where*

$$\widetilde{T}(\gamma; g) = \inf \left\{ t \colon \frac{1}{n+1} \left( B + \sum_{i=1}^{n} g(L_i(t)) \right) \leq \gamma \right\}. \tag{17}$$

Following their analysis, we give a second corollary for lower bounding function $R_{n+1}$, where $R_1, \ldots, R_{n+1}$ are now non-decreasing functions.

**Corollary A.3** (Marginal RCPS, monotonic non-decreasing case). *Similar to the setting in Theorem A.1, let $R_i \colon \mathbb{R} \to \mathbb{R}$, $i = 1, \ldots, n+1$ be exchangeable functions, where $R_i(t)$ is non-decreasing in $t$. Also, take $g \colon \mathbb{R} \to \mathbb{R}$ where $g(x)$ is non-decreasing in $x$. Further assume that $g \circ R_i$ is right-continuous, and*

$$\inf_t g(R_i(t)) \geq 0, \quad \sup_t g(R_i(t)) > C \geq \gamma \text{ almost surely.} \tag{18}$$

*For any $\gamma \leq 0$, define the random variable $T(\gamma, g)$ as*

$$T(\gamma; g) := \inf \left\{ t \colon \frac{1}{n+1} \sum_{i=1}^{n} g(R_i(t)) \geq \gamma \right\}, \tag{19}$$

*where we define $\inf \varnothing = \infty$. Then $\mathbb{E}[g \circ R_{n+1}(T(\gamma; g))] \geq \gamma$.*

*Proof.* Let

$$T'(\gamma; g) := \inf\left\{t\colon \frac{1}{n+1}\sum_{i=1}^{n+1} g(R_i(t)) \geq \gamma\right\}. \tag{20}$$

Since $\inf_t g(R_i(t)) \geq 0$, $\sup_t g(R_i(t)) > C \geq \gamma$, $T'(\gamma; g)$ and $T(\gamma, g)$ are both well-defined almost surely. Since $\inf_t g(R_i(t)) \geq 0$,

$$\frac{1}{n+1}\sum_{i=1}^{n} g(R_i(t)) \geq \gamma \rightarrow \frac{1}{n+1}\sum_{i=1}^{n+1} g(R_i(t)) \geq \gamma. \tag{21}$$

Thus, $T'(\gamma; g) \leq T(\gamma; g)$. Since $g(R_i(t))$ is non-decreasing in $t$,

$$\mathbb{E}[g \circ R_{n+1}(T(\gamma; g))] \geq \mathbb{E}[g \circ R_{n+1}(T'(\gamma; g))]. \tag{22}$$

Let $E_f$ be the unordered set (bag) of $\{R_1, \ldots, R_{n+1}\}$. Then $T'(\gamma; g)$ is a function of $E_f$, and is a constant conditional on $E_f$. Exchangeability of $R_i$ and right-continuity of $g \circ R_i$ imply

$$\mathbb{E}[g \circ R_{n+1}(T'(\gamma; g)) \mid E_f] = \frac{1}{n+1}\sum_{i=1}^{n+1} g \circ R_i(T'(\gamma; g)) \geq \gamma. \tag{23}$$

The proof is completed by taking the expectation over $E_f$ and then applying Eq. (22). $\qquad\square$

We now prove our main theorem.

*Proof.* W.l.o.g. let us the flip signs of our non-conformity measures such that

$$\overline{\mathrm{MFP}}(x, \mathcal{Z}, t) := \max\left\{\mathrm{FP}(\mathcal{Z}, \mathcal{S}_i)\colon -\widetilde{\mathcal{F}}(x, \mathcal{S}_i) \geq t\right\}, \tag{24}$$

and $T_k$, $T_{k,\delta}$ are instead defined via $\inf$.

We are given that the sets $\mathcal{S}$ are nested, and it is clear from its definition that FP is non-decreasing for $\mathcal{S}_i \subset \mathcal{S}_j$. Since the ranking of $-\widetilde{\mathcal{F}}(x, \mathcal{S})$ is consistent with that of $-|\mathcal{S}|$ and monotonically non-increasing, $\overline{\mathrm{MFP}}$ is therefore also monotonically non-increasing in $t$.

We prove $T_k$ first.

Since $B$ is constrained to be finite, we have that $\sup_t \overline{\mathrm{MFP}}(x, \mathcal{Z}, t) \leq B < \infty$. Furthermore, since $-\widetilde{\mathcal{F}}(X_i, S_{i,j})$ is upper-bounded by assumption, we have that $\inf_t \overline{\mathrm{MFP}}(x, \mathcal{Z}, t) = 0 < k \in \mathbb{R}_{>0}$. Finally, as evident from its definition in Eq. (24), $\overline{\mathrm{MFP}}(x, \mathcal{Z}, t)$ is right-continuous. Let $L_i(t) = \overline{\mathrm{MFP}}(X_i, Z_i, t)$ and $g(x) = x$. We can then directly apply Corollary A.2 to obtain

$$\mathbb{E}[\mathrm{MFP}(X_{n+1}, Z_{n+1}, T_k)] \leq k. \tag{25}$$

The proof for $T_k$ is completed by the fact that $\mathrm{MFP}(X_{n+1}, Z_{n+1}, T_k) \geq \mathrm{FP}(Z_{n+1}, \mathcal{S}_{n+1,j})$ conditioned on $\widetilde{\mathcal{F}}(X_{n+1}, \mathcal{S}_{n+1,j}) \leq T_{k,\delta}$.

We proceed similarly for $T_{k,\delta}$.

Let $L_i(t) = \mathbf{1}\{\overline{\mathrm{MFP}}(x, \mathcal{Z}, t) \leq k\}$. Since $\overline{\mathrm{MFP}}$ is non-increasing and right-continuous, $L_i(t)$ is non-decreasing and right-continuous. Let $\gamma = 1 - \delta \in (0, 1)$. By the same argument as before concerning $\overline{\mathrm{MFP}}$, and from the fact that $L_i(t) \in \{0, 1\}$, we can apply Corollary A.3 to get

$$\mathbb{E}[\mathbf{1}\{\mathrm{MFP}(X_{n+1}, Z_{n+1}, T_{k,\delta}) \leq k\}] = \mathbb{P}(\mathrm{MFP}(X_{n+1}, Z_{n+1}, T_{k,\delta}) \leq k) \geq 1 - \delta. \tag{26}$$

This gives $\mathbb{P}(\mathrm{MFP}(X_{n+1}, Z_{n+1}, T_{k,\delta}) \leq k) \geq 1 - \delta$. The proof for $T_{k,\delta}$ is then completed by the fact that $\mathrm{MFP}(X_{n+1}, Z_{n+1}, T_{k,\delta}) \geq \mathrm{FP}(Z_{n+1}, \mathcal{S}_{n+1,j})$ conditioned on $\widetilde{\mathcal{F}}(X_{n+1}, \mathcal{S}_{n+1,j}) \leq T_{k,\delta}$. $\quad\square$

## A.3 PROOF OF PROPOSITION 4.6

*Proof.* By definition, we have $\mathcal{F}(X_{n+1}, \mathcal{C}_\circ(X_{n+1}, T_\circ)) \leq T_\circ$, where $\circ$ is a placeholder for $k$ or $k, \delta$. By Theorem 4.3, we can then conclude that

$$\mathbb{E}[\text{FP}(Z_{n+1}, \mathcal{C}_k(X_{n+1}, T_k))] \leq k, \tag{27}$$

and

$$\mathbb{P}(\text{FP}(Z_{n+1}, \mathcal{C}_{k,\delta}(X_{n+1}, T_{k,\delta})) \leq k) \leq T_{k,\delta}) \geq 1 - \delta. \tag{28}$$

Finally, if $\mathcal{S} \subset \mathcal{S}'$ then $y_c \in \mathcal{S} \Rightarrow y_c \in \mathcal{S}'$ for any $y_c \in z \subseteq \mathcal{Y}$. Since the candidate sets are nested,

$$|\mathcal{S}| < |\mathcal{S}'| \implies \text{TPP}(z, \mathcal{S}) \leq \text{TPP}(z, \mathcal{S}') \tag{29}$$

Thus picking the biggest candidate set maximizes the TPP for a given $Z = z$, as well as its expectation over random $Z$, i.e., the TPR. □

## B CONFORMAL COVERAGE RESULTS

| Task | TPR | Avg. FP | Avg. Size |
|---|---|---|---|
| In-silico screening | 97.2 | 63.6 | 86.6 |
| Object detection | 96.1 | 32.4 | 38.2 |
| Entity extraction | 75.0 | 0.77 | 2.31 |

Table B.1: Outer Sets results applied at coverage level $1 - \epsilon = 0.90$. Note that since some examples do *not* have any positives, full coverage in the typical sense isn't always achievable.

## C IMPLEMENTATION AND DATASET DETAILS

| Dataset | Input | # Examples | # Negatives | # Positives | % Empty |
|---|---|---|---|---|---|
| In-silico screening | SMILES | 5,000 | 85 (50-97) | 15 (3-50) | 0.0 |
| Object detection | Image | 3,000 | 96 (89-98) | 4 (2-11) | 1.1 |
| Entity extraction | Text | 3,453 | 99 (97-100) | 1 (0-3) | 20.2 |

Table C.1: Dataset statistics (test split). Numbers are reported with respect to the top $B = 100$ candidates per example. The median number of positives and negatives per example is given, in addition to their 16th and 84th percentiles. Examples with no positives ($|z| = 0$) are treated as empty.

**In-silico screening.** We construct a molecular property screening task using the ChEMBL dataset (see Mayr et al., 2018). Given a specified constraint such as "*is active for property A but not property B*," we want to retrieve at least one molecule from a given set of candidates that satisfies this constraint. The input for each molecule is its SMILES string, a notational format that specifies its molecular structure. The motivation of this task is to simulate in-silico screening for drug discovery, where it is often the case where chemists are searching for a new molecule that satisfies several constraints (such as toxicity and efficacy limits), out of a pool of many possible candidates.

We split the ChEMBL dataset into a 60-20-20 split of molecules, where 60% of molecules are separated into a train set, 20% into a validation set, and 20% into a test set. Next, we take all properties that have at least 50 positive and negative examples (to avoid highly imbalanced properties). Of these properties, we take all N choose K combinations that have at least 100 molecules with all K properties labelled (ChEMBL has many missing values). We set K to 2. For each combination, we randomly sample an assignment for each property (i.e., $\{\text{active}, \text{inactive}\}^K$). We discard combinations for which more than 90% of labeled molecules satisfy the constraint. We keep 5000 combinations for the test set, 767 for validation, and 4375 for training. The molecules for each of the combinations are only sourced from their respective splits (i.e., molecular candidates for constraints in the property combination validation split only come from the molecule validation split). Therefore, at inference

time, given a combination we have never seen before, on a molecules we have never seen before, we must try to retrieve at least one molecule that has the desired combination assignment.

Our directed Message Passing Neural Network (MPNN) is implemented using the `chemprop` repository (Yang et al., 2019). The MPNN model uses graph convolutions to learn a deep molecular representation, that is shared across property predictions. Each property value (active/inactive) is predicted using an independent classifier head. The final prediction is based on an ensemble of 5 models, trained with different random seeds. Given a combination assignment $(Z_1 = z_1, \ldots, Z_k = z_k)$, we compute the joint likelihood independently, i.e.,

$$p_\theta(Z_1 = z_1, \ldots, Z_k = z_k) = \prod p_\theta(Z_i = z_i).$$

**Object detection.** As discussed in §5, we use the MS-COCO dataset (Lin et al., 2014) to evaluate our conformal object detection. MS-COCO consists of images of complex everyday scenes containing 80 object categories (such as person, bicycle, dog, car, etc.), multiple of which may be contained in any given example. Since the official test set is hidden, we use the $5k$ examples from the development set and randomly partition them into sets of size $1k$, $1k$, and $3k$ for calibration, validation, and testing, respectively. The EfficientDet model (Tan et al., 2020)[4] for extracting bounding boxes uses a pipeline of three neural networks to extract deep features, and then predict candidates. The model also uses a non-maximum suppression (NMS) post-processing step to reduce the total number of predictions by keeping only the one with the maximum score across highly overlapping prediction boxes. We merge the predictions of all classes into a unified set, where each element is a tuple of (class, bounding box). This means that multiple class predictions can be included for the same bounding box (i.e., there is class uncertainty), and multiple bounding boxes can be found for the same class (i.e., there are multiple objects in one image). We define true positives as predictions that have an intersection over union (IoU) value $> 0.5$ with a gold bounding box annotation, *and* that match the annotation's class.

**Entity extraction.** Entity extraction is a popular task in natural language processing. Given a sentence, such as "*Barack Obama was born in Hawaii*," the goal is to identify and classify all named entities that appear—i.e., ("Barack Obama", Person) and ("Hawaii", Location). We use the CoNLL NER dataset (Tjong Kim Sang and De Meulder, 2003), and extract $1k$ examples for calibration out of the $3.3k$ development set, and report test results on the $3.5k$ test set. For our base model, we use the entity extraction module of PURE (Zhong and Chen, 2021), that predicts span scores with a classifier head on top of Albert-base (Lan et al., 2020) contextual embeddings. The classification head has two non-linear layers and uses the learned contextual embeddings of the span start and end tokens, concatenated with a learned span width embedding. We train the model on the training set of the CoNLL NER dataset. We use the official code repository[5] and the following parameters: $1e - 5$ learning rate, $5e - 4$ task learning rate, 32 train batch size, and 100 context window. Similar to our object detection task, we treat exact span predictions of the correct category as true positives, and any other entity predictions as false positives. As illustrated in Table C.1, a fairly large number of sentences do not contain any entities at all, while other sentences may contain several.

## D  PRACTICAL CONSIDERATIONS

In this section we address a number of practical considerations for our FP-CP method.

### D.1  CHOOSING A SUITABLE $k$

An outstanding question a practitioner faces is how to choose the value of $k$ for $k$-FP and $(k, \delta)$-FP objectives. The value of $k$ in our method has a reliable and easy interpretation: it is the total number of incorrect answers. For many tasks, such as in-silico screening, there is a direct relation between the number of noisy predictions (e.g., failed experiments conducted during wet-lab validation) and total "wasted" cost. Therefore, for example, given some approximate budget $Q$ and cost per noisy prediction $c$, a reasonable approach is to then set $k \approx Q/c$. Of course, the advantage of our approach is that the user may set $k$ to whatever they wish—this might change based on their needs.

### D.2  CHOOSING BETWEEN $k$-FWER AND FDR CONTROL

A related question to D.1 is when to target $k$-FWER (i.e., our $k$-FP and $(k, \delta)$-FP objectives) or FDR (e.g., using Angelopoulos et al. (2021a)). This choice is well discussed in the multilple testing

---

[4]We use tf_efficientdet_d2 from `https://github.com/rwightman/efficientdet-pytorch`.
[5]`https://github.com/princeton-nlp/PURE`.

literature (Lehmann and Romano, 2005; Romano and Wolf, 2007; Gold et al., 2009). An important aspect to consider is the size of the label space $\mathcal{Y}$, natural rate of true and false positives, and the efficiency of the base model at separating true positives from false positives. When the total number of true positives is large and $|\mathcal{Y}|$ is large then it is reasonable to control the FDR. If, however, the natural rate of true positives is low, or they are not well separated from false positives, then the FDR can be quite high and hard to control, especially for smaller prediction sets (as the ratio of positive to negative labels can be quickly driven down even with the addition of only a few false positives). As a intuition illustration, suppose for a given example there is one true positive that is ranked 10th by the base model. For many applications, 10 total predictions (with 9 false positives) is not overly burdensome. However, the lowest FDR cutoff that allows for this positive to be discoverable is 0.9 (which, for other examples, may allow for hundreds of false positives—an outcome which may not be so desirable for some applications, even given a high number of accompanying true positives). To satisfy a lower FDR rate, the algorithm must output an empty set (with FDR = 0). This remains true even if there are a few (but not many) other true positives: for instance, in the previous example, if predictions 10-20 were also all true positives then the lowest FDR is still only 0.5—specifying a FDR tolerances any lower than this would force an empty set prediction.

### D.3  LEARNING MORE EXPRESSIVE SET FUNCTIONS

Our choice of $\mathrm{DeepSets}$ model is motivated by its property of being a universal approximator for continuous set functions. Of course, its realized accuracy depends on its exact parameterization and optimization. In terms of input features, in §4.2, we chose a simplistic $\phi(x, y_c)$ for two reasons: (1) we view it's simplicity as an advantage (practitioners can simply plug-in individual multi-label probabilities, or other scalar conformity scores, that most out-of-the-box methods provide into a general framework without having to do any more work for providing additional features), and (2) it is easy to train this light-weight model on smaller amounts of data. This approach, however, can discard potentially helpful information about the underlying input $x$, and any dependencies between labels $y_c$ and $y_c'$. For example, if $y_c$ and $y_c'$ are mutually exclusive, then the number of false positives if both are included in $\mathcal{S}$ has to be at least 1. Using more expressive $\phi$ that is able to capture and take advantage of this sort of side information about $x$ and $y_c$ is a subject for future work.

### D.4  CONSTRUCTING NON-NESTED CANDIDATE SETS

Our careful construction of nested prediction sets that have bounded, non-decreasing, and right-continuous risks is key to our calibration procedure. It is, however, limited by the underlying ranking of labels $y_c \in \mathcal{Y}$ by the individual likelihood model $p_\theta(y_c \in Z \mid x)$. In theory, the set function $\mathcal{F}$ may be able to identify higher quality outputs sets $\mathcal{S} \in 2^{\mathcal{Y}}$ by jointly considering all of the included elements (rather than ranking them one-by-one). That said, the search process over $2^{\mathcal{Y}}$ is expensive, and the FP risk is no longer non-decreasing, which necessitates more involved calibration methods that can account for non-monotonic risks, such as that of Angelopoulos et al. (2021a)). Nevertheless, this is a promising subject for future work, and one that can potentially be combined with efficient search or candidate space pruning methods (e.g., such as in Fisch et al. (2021)).

### D.5  FAILURE CASES OF GREEDY OUTPUT SELECTION

In §4.4 we greedily select the largest candidate set that passes our filter threshold. A downside of this approach is that it may include more false positives than necessary (i.e., to achieve the same TPR). Ideally, we would be able to follow the oracle strategy in returning the smallest set with the highest true positive proportion. This would make our predictions *efficient*, in the sense that we are not including more false positives than necessary (even if the total number is still $\leq k$). A reasonable choice is to then choose $\mathcal{S}_{j^*}$ where $j^* = \arg\max_j |\mathcal{S}_j| - \mathcal{F}(x, \mathcal{S}_j)$; but this can be sub-optimal if $\mathcal{F}$ is not accurate, which motivates our greedy strategy. Nevertheless, non-greedy set selection may be more efficient if $\mathcal{F}$ is indeed accurate—this can be tested on a validation set.

