# OpenReview forum: "Trading Coverage for Precision: Conformal Prediction with Limited False Discoveries"
_ICLR.cc/2022/Conference — ICLR 2022 Submitted_

### Official Review · Reviewer_yMiK · 2021-10-17

**Correctness:** 3
**Technical Novelty And Significance:** 3
**Empirical Novelty And Significance:** 3
**Recommendation:** 6
**Confidence:** 4

**Main Review:**

Overall, the paper is well-written and I enjoyed reading it. I believe it is a good contribution to the ever-growing literature on conformal inference. Below, I entail several comments about the proposed method.

The first comment is about the apriori choice of k in k-FWER. How can the user pre-specify this value? This is one of the limitations of using this proposed error rate compared to the choice of FDR. Currently, the paper does not discuss this issue, which I believe is central for deploying the proposed technique in real-world applications.

Still in that context, can you explain what is the outstanding theoretical challenge in proving the k-FWER control compared to the FDR-based procedure suggested in the original RCPS paper?

I understand that forming $\phi(x,y_i)$ as the marginal likelihood of $y_i$ being a correct label for $x$ is simple to implement. However, this seems to be suboptimal. To me, one of the major contributions of this work is the formulation of the deep net model to estimate the distribution of $S$, and therefore it will be a plus to explore other methods for estimating $\phi(x,y_i)$ beyond this simplistic approach (or, at least, discuss other options). Similarly, it will be interesting to understand when the greedy approach to identifying a sequence of nested candidate sets (described on page 5 bottom to page 6 top) will fail. Here, a synthetic simulation can be very useful.

As for $\Psi$ (eq. 8), the authors stated that they use labeled sets sampled from auxiliary training data. Can you elaborate more on that?

Regarding Remark 4.4: can you provide a small synthetic experiment that illustrates this issue? Also, when |\mathcal{Y}| is large, it seems to me that using the FDR control procedure is more natural and can be more powerful (as long as the number of true positives is large).

Experiments: the authors mentioned (Section 2) that “[...] as most of our target applications have relatively few true positives, FDR control can be somewhat volatile and lead to many empty predictions.” I believe it will be valuable to demonstrate this issue in experiments, to support the need for k-FWER control.

Lastly, please include in the Supplementary Material more details on the data sets used in the experiments.


**Summary Of The Paper:**

This paper presents a new calibration procedure that builds upon the framework of Distribution free, risk-controlling prediction sets (RCPS) by Bates et al. (2020). The latter trades off the coverage guarantees of conformal prediction in favor of precision guarantees. While in RCPS the authors of RCPS show how to control the rate of false positives (false discovery rate, FDR), this paper presents a modification that puts a limit on the number of false positives that are contained in the constructed prediction sets. In other words, here the focus is on controlling the k family-wise error rate (k-FWER) instead of the FDR. The second contribution is the formulation of a deep net model that estimates the distribution of false positives in a candidate prediction set. This model is plugged into the proposed calibration scheme, used to find a threshold for rejecting or accepting candidate sets generated by the model. Lastly, in a series of experiments, the authors confirm the validity of the theoretical guarantees presented in the paper as well as the advantage of the proposed deep net model over baseline methods.

**Summary Of The Review:**

I believe this paper can be improved by addressing the above comments, e.g., how to pre-specify k (if possible), discussing alternative options to build the nested sets (that may be more complex), and explaining possible failure cases of the greedy approach. Also, discussing (e.g., via simulated data) the advantages and limitations of k-FWER over FDR seem to be important for the users to better understand which error metric is preferred, given the nature of the data at hand (and the existence of prior work).

---

> ### Author Response · Authors · 2021-11-12
> **Response**
>
> We thank the reviewer for their comments and suggestions. We are committed to improving our paper and can easily address all concerns within the rebuttal period. Below, we address the reviewer’s comments and indicate in **bold** additional experiments or updates to the writing that we will post within several days. We hope that these will allay any concerns about our work and convince the reviewer that it will be a welcome contribution to the ICLR community. **If there are additional experiments or clarifications that we can perform to further support our case, please let us know.**
>
> > _The first comment is about the apriori choice of k in k-FWER. How can the user pre-specify this value? This is one of the limitations of using this proposed error rate compared to the choice of FDR._
>
> Please see our response to reviewer dWtw, who had a similar question. In general, for the tasks that we consider, it is easier to reason about the absolute cost of false positives rather than their ratio: for example, in the case of in-silico screening, there is a direct relation between # of failed experiments during validation and total “wasted” cost. As a result, k can be set relative to what the user's needs and budget are.
>
> Other benefits of k-FWER control over FDR control are due to what we mention (as you also note) in Sec. 2: that FDR makes sense most when the total number of both positives and negatives are high. In most of our experiments, the number of true positives is low. Of course, when these assumptions change, FDR could certainly be a viable objective. We believe that both k-FWER and FDR formulations are valuable and complementary.  **We will add additional synthetic experiments to demonstrate this.**
>
> > _Can you explain what is the outstanding theoretical challenge in proving the k-FWER control compared to the FDR-based procedure suggested in the original RCPS paper?_
>
> Note that FDR is a non-monotonic risk, and was only very recently covered in Angelopoulos et. al., 2021a (a followup to RCPS). Our procedure also provides a simpler form of marginal control, and builds on the very recent results by Angelopoulos et. al., 2021b (marginal RCPS, another follow-up). The theoretical reframing for our particular formulation is non-trivial, and requires corollary A.3 as well as a careful construction of prediction sets that have bounded, non-decreasing, and right-continuous risks. That said, we do fully acknowledge the foundational results that enable our rigorous procedure---both in the introduction and related work.
>
> > _To me, one of the major contributions of this work is the formulation of the deep net model to estimate the distribution of S, and therefore it will be a plus to explore other methods for estimating $\phi(x, y_i)$ beyond this simplistic approach (or, at least, discuss other options)._
>
> We chose a simplistic $\phi$ for two reasons: (1) we view it’s simplicity as an advantage (practitioners can simply plug-in individual multi-label probabilities, or other scalar conformity scores, that most out of the box methods provide into a general framework without having to do any more work for providing additional features), and (2) it is easy to train this light-weight model on smaller amounts of data. This goes to your next comment: the auxiliary data used to train $\Psi$ is split off from the original training data (see Alg 1, line 6)---though other, more data efficient, options are also possible (see our response to reviewer 2ZTR).
>
> That said, using more expressive $\phi$ (that takes advantage of more side information, and better models dependencies between labels) is certainly possible. Full consideration of this can be rather application specific, however, and merits future work (more than the scope of this work).  **We will update the manuscript to address this comment.**

---

> > ### Comment · Reviewer_yMiK · 2021-11-25
> > **Follow-up**
> >
> > Thank you for the response and clarifications.
> >
> > *Regarding your answer to the comment raised by Reviewer 2ZTR: **TDR appears to asymptote at 0.8**.* I would argue that, in this case, the best output should be an empty set rather than a set that merely contains false positives. This is tightly connected to the abstain mechanism presented recently in [1]. It will be a plus if you could comment on that.
> >
> > [1] Anastasios N. Angelopoulos, Stephen Bates, Emmanuel J. Candès, Michael I. Jordan, Lihua Lei, "Learn then Test: Calibrating Predictive Algorithms to Achieve Risk Control," arXiv:2110.01052, 2021.

---

> > > ### Author Response · Authors · 2021-11-25
> > > **Response to follow-up**
> > >
> > > Thank you for the follow-up question.
> > >
> > > We address this point in Appendix D.5. Indeed, this is a feature of our _greedy_ set selection procedure. Without knowing how many true positives there actually are for our task, we try to do the best we can within our respective false positive budget. Returning the largest set that passes this test maximizes our TPR rate.
> > >
> > > Of course, as you suggest, and as we note in Appendix D.5, this may be inefficient (not just for examples with no true positives, but whenever we output a set that is larger than necessary for the same TPR). As we state, however, this is not a known quantity, but rather one we have to estimate (and when we estimate it, we may be wrong). So our greedy procedure is a simple, and robust, way of proceeding. Nevertheless, as we describe in Section 4.4, we are free to select any set in the filtered set of candidates that pass our set-score threshold (which includes $\emptyset$). So, for example, one could make a decision rule that paired our procedure with a standard abstention-based classifier $g$ if this behavior was important (i.e., predict the greedy set if $g(x_{n+1}) = 1$ and $\emptyset$ otherwise). And this wouldn't require any extra calibration (in the conformal, FP constraint preserving sense).
> > >
> > > The abstention-based procedure of Angelopoulos et. al. that you reference is indeed interesting and connected to this particular setting. However, it is based on bounding the pFDR, and shares the same conservative aspects as FDR-control that we discussed in our earlier response to your review and Appendix D.2 in our updated manuscript.

---

> ### Author Response · Authors · 2021-11-24
> **Response and updated manuscript**
>
> Dear Reviewer yMiK,
>
> Thank you once again for your time and helpful comments and suggestions. As posted in the general response, we have submitted a revised version of our paper that we believe addresses the questions that you raised. If you have any further comments, please do let us know.

---

### Official Review · Reviewer_dWtw · 2021-11-01

**Correctness:** 3
**Technical Novelty And Significance:** 3
**Empirical Novelty And Significance:** 3
**Recommendation:** 6
**Confidence:** 3

**Main Review:**

In general, the paper is well-written, with a detailed review of relevant prior works in the literature. It focuses on uncertainty quantification for settings where certain budget constraints are imposed by a particular application, e.g., when adding an extra element to a prediction set has to be subsequently verified experimentally. The approach roughly boils down to: (a) fitting a multi-label predictor, (b) fitting a model that estimates the number of false discoveries, (c) using a proposed non-conformity score and a held-out set for calibrating a final threshold (subsequently used for constructing prediction sets) that allows provably controlling the number of false discoveries.

--- I was wondering whether the authors could elaborate more on the advantages of controlling the total number of false discoveries when compared against controlling its normalized version, false discovery rate, for which actually results have been established recently. The question arises mainly due to choosing $k$ in practice. While controlling FDR at level 0.1 is easy to interpret, what happens when one considers the total number of false discoveries?

--- A side notes about some typos:
a. Several typos in Theorem 3.1: in equation 3, $x_{n+1}$ should be replaced with $x$. Also, in the line that follows equation 3, $C_\epsilon(X_{n+})$ should be replaced with $C_\epsilon(X_{n+1})$.

**Summary Of The Paper:**

Being capable of quantifying uncertainty is critical in various applications of ML systems. Conformal prediction is a framework that allows building a wrapper around a predictive model. Subsequently, a point prediction (corresponding to the top-ranked label) is replaced by outputting a set of labels that provably contains the true label of a test point with high probability (with the guarantees being only marginal). In particular, split conformal prediction proceeds by scoring every possible label of a test point and comparing these scores against the ones computed on a held-out set, followed by the decision of whether a particular label has to be included in the prediction set. However, in many applications due to present noise, the resulting prediction sets are larger / more conservative than expected, with many incorrect labels being included, and the presence of many false positives could be an unfavorable event in many real-world scenarios from the actionability standpoint (e.g., drug discovery, where adding an extra element to the prediction set corresponds to a significant cost increase of an experiment). The current paper studies a way of trading off coverage for precision, from both theoretical and practical standpoints.

**Summary Of The Review:**

The current paper represents a solid piece of work. The main reason for lowering the score is that several papers have considered post-hoc uncertainty quantification procedures for multi-label classification settings (possibly a more detailed lit review with focus on this is necessary). The authors propose a way of controlling the total number of false discoveries (either in expectation or with high probability) based on conformal inference. The score could be updated once the questions mentioned in the main review are answered.

**Update after rebuttal**

I would like to thank the authors for the detailed responses. Taking into account the general contribution of this work and points mentioned in this and other reviews, I tend to keep the current score.

---

> ### Author Response · Authors · 2021-11-12
> **Response**
>
> We thank the reviewer for their comments and suggestions. We are committed to improving our paper and can easily address all concerns within the rebuttal period. Below, we address the reviewer’s comments and indicate in **bold** additional experiments or updates to the writing that we will post within several days. We hope that these will allay any concerns about our work and convince the reviewer that it will be a welcome contribution to the ICLR community. **If there are additional experiments or clarifications that we can perform to further support our case, please let us know.**
>
> > _I was wondering whether the authors could elaborate more on the advantages of controlling the total number of false discoveries when compared against controlling its normalized version, false discovery rate, for which results have been established recently._
>
> This is a good question. In very recent work (around the time of this submission), Angelopoulos et. al. 2021a proposes general non-monotonic risk controlling procedures that can be used to control the FDR, as we mention in our related work section. As we explained in that section as well, when the number of true positives is small, the FDR can be hard to control. For example, if you have a correct answer in your top 3 predictions---most practitioners would consider this to be good! However, this has a FDR of 0.66, which would also allow for predictions that have hundreds of false positives (not so good). In general, for the tasks that we consider, it is easier to reason about the absolute cost of false positives rather than their ratio: for example, in the case of in-silico screening, there is a direct relation between # of failed experiments during validation and total “wasted” cost. We argue that both formulations (FDR and the k-FWER discussed in this work) are valuable, and complementary, contributions. **We will discuss this choice more directly in the manuscript.**
>
> > _Several typos in Theorem 3.1_
>
> Thank you for pointing this out. **We will update this in the manuscript.**

---

> ### Author Response · Authors · 2021-11-24
> **Response and updated manuscript**
>
> Dear Reviewer dWtw,
>
> Thank you once again for your time and helpful comments and suggestions. As posted in the general response, we have submitted a revised version of our paper that we believe addresses the questions that you raised. If you have any further comments, please do let us know.

---

### Official Review · Reviewer_KRhW · 2021-11-03

**Correctness:** 4
**Technical Novelty And Significance:** 2
**Empirical Novelty And Significance:** Not applicable
**Recommendation:** 6
**Confidence:** 4

**Main Review:**

## Strengths

The method is well-motivated from a practical standpoint. It is also well-grounded in theory without burdensome assumptions.

## Weaknesses

The main issue has to do with clarity. The writing is difficult to follow, almost always unnecessarily so.

1. *Inconsistent or ill-defined notations*: The following is not meant to be an exhaustive list.

- On p. 1, $(X_i, Z_i) \in \mathcal{X} \times 2^{\mathcal{Y}}$, but $\mathcal{Y}$ has not been defined.
- Both $Z_{n+1}$ and $\mathcal{Z}_{n+1}$ as used. As far as I can see, they represent the same quantity.
- Similarly, both $S_{i}$ and $\mathcal{S}_{i}$ are in use.
- $B$ in Algorithm 1 is a hyper-parameter that caps the number of different values of $y$ that is considered. $B$ on p. 6 is a bound on the cardinality of $|\mathcal{Y}|$. The two definitions are in conflict. There is also $m$ in Remark 4.4, which appears to have the same meaning as the first $B$.
- It is confusing to have the same index $i$ be used to index many different quantities. It is initially introduced as indexing observations, e.g., $(X_i, Z_i)$. It is next used to index candidate labels, e.g., $\phi(x, y_i)$. It is again used to index sets in a nested sequence of sets, e.g., $\mathcal{S}_{i}$.
- The situation is made worse when there is a need for double-indexing, e.g., $\mathcal{S}_{i, j}$. All this is easily avoidable by introducing a different index whenever the meaning changes.
- The same comment applies to the remark about $k$ in Top-$k$.
- $t^*$ and $T_*$ appear to have the same meaning.
- Is $\operatorname{SSFD}_k$ in Eq. (14) the same as $\operatorname{WCSS}$ in Figures 2 and 3?

2. I do not think the paper ever explicitly spells out the precise meta-algorithm (the FD-CP algorithm) for either $k$-FD validity or $(k, \delta)$-FDP validity that can represent the proposed method. Algorithm 1 is only a particular *instance* of the proposed method that relies on specific choices of learners (MPNN and DeepSets) for the problem of in-silico screening. On a related note, the discussion in Section 4 does not appear to sufficiently differentiate between the general construction (e.g., nonconformity scores for nested sequences of sets) versus a particular example (e.g., the scores of Eq. (9)).

3. In Section 5, four baseline methods are described, but the reason behind the choice is never explicitly given. In particular, one of the baseline methods is not expected to have validity, raising the obvious question of why the method was considered in the first place. The last two (max and avg) were probably included as alternative scoring methods to the scoring method using DeepSets, but this is an educated guess. Also, if this was indeed the intention, then I feel like more space should have been devoted to a discussion of what makes DeepSets an appropriate choice, especially in light of Remark 4.5.

4. Both figures are hard to read. The plotting areas are too small, the lines overlap too much, and the colors are too similar. This appears to be due in part to imposing the same scale on the $y$-axes. Also, the top and the bottom panels are misaligned.

5. Overall, I feel like the current version of the manuscript does the bare minimum when it comes to explaining the intuition underlying the approach.

## Minor
- (p. 1. The 1st line of Section 1) instead of single prediction -> instead of a single prediction
- (p. 2. The 4th line after Eq. (2)) Spell out the initialisms.
- (p. 5, The 1st line after Eq. (8)) What are enc and dec?
- (p. 6. Theorem 4.3) $\tilde{\mathcal{F}}(X_{n+1}, \mathcal{S}_{n+1, j}) \leq T_k$ or $T_{k, \delta}$, rather?
- (p. 7. **In-silico screening for drug discovery**) I think the correct usage places the comma before the end quote and not after.
- (p. 7. **Entity extraction**) The letter $l$ has already been used earlier in the text.
- (p. 9. The 2nd line of Section 7) can be a critical in -> can be critical in
- (Appendix) Please check the punctuations in display equations.
- (p. 14) In Eq. (24), is $t$ in the middle expression $T'(\gamma; g)$? Do we take the expectation over $E_f$ before using Eq. (23)?
- (p. 14) w.l.o.g. -> W.l.o.g.
- (p. 14. The 2nd line after Eq. (26)) $\mathcal{S}_{n+1, j} \leq T_k$ -> $\tilde \mathcal{F} (X_{n+1}, \mathcal{S}_{n+1, j}) \leq T_k$
- (p. 14. The last line of the proof) $\mathcal{S}_{n+1, j} \leq T_{k, \delta}$ -> $\tilde \mathcal{F} (X_{n+1}, \mathcal{S}_{n+1, j}) \leq T_{k, \delta}$.

## Additional Comments

I am not sure if the title "Trading Coverage for Precision" is a good fit for the method being proposed. I would say that the method is more about using achieving a different target for validity. It looks to me whether the effect is to produce more precise prediction sets would depend on the dataset as well as the model being used.

**Summary Of The Paper:**

A new conformal method for multi-class prediction problems is proposed in which the number of false discoveries is capped while the proportion of true discoveries is maximized. The approach is shown to achieve the desired control in finite samples. The claim is further supported by experimental results.

**Summary Of The Review:**

As of writing, I am on the fence about whether to recommend this paper for acceptance or rejection. On the one hand, the underlying ideas do appear to be sound and of merit. However, the issues of clarity are such that I cannot be sure of having evaluated them correctly.

---

> ### Author Response · Authors · 2021-11-12
> **Response**
>
> We thank the reviewer for their comments and suggestions. We are committed to improving our paper and can easily address all concerns within the rebuttal period. Below, we address the reviewer’s comments and indicate in **bold** additional experiments or updates to the writing that we will post within several days. We hope that these will allay any concerns about our work and convince the reviewer that it will be a welcome contribution to the ICLR community. **If there are additional experiments or clarifications that we can perform to further support our case, please let us know.**
>
> > _Inconsistent or ill-defined notations_
>
> Thank you for pointing these out. In particular, we agree with the reviewer that the usage of B is confusing; see also our response to reviewer 2ZTR. WCSS is a typo in Fig. 2 and 3: it is meant to be the same as SSFD_k. **We will address all of these issues in an update to the manuscript. **
>
> > _I do not think the paper ever explicitly spells out the precise meta-algorithm (the FD-CP algorithm) for either k-FD validity or (k,δ)-FDP validity that can represent the proposed method. Algorithm 1 is only a particular instance of the proposed method that relies on specific choices of learners (MPNN and DeepSets) for the problem of in-silico screening._
>
> We tried to strike a balance between presenting an abstract procedure and theoretical results, and one that was grounded in instantiated models that are easier to immediately follow for a practitioner. **We will update the manuscript to include a precise statement of the general-case meta-algorithm.**
>
> > _In Section 5, four baseline methods are described, but the reason behind the choice is never explicitly given._
>
> Our motivation is to show how our proposed method compares to simpler ones. Even though top-k (without more careful calibration) isn’t guaranteed to be valid, it is still an obvious (but suboptimal) choice that a practitioner might make. We wanted to compare directly to these results, even though our method has the added (strict) validity constraint. For a discussion of why deep sets is an appropriate model, it is a simple, direct instantiation of a valid set function. Thm. 2 of the deep sets paper, states that a function $f(X)$ operating on a set $X$ is a valid set function iff it can be decomposed in the form $\rho(\sum_{x \in X} \phi(x))$ for suitable $\rho$ and $\phi$.  They also empirically find that this direct form works well. We address this in our response to reviewer 2ZTR as well.
>
> > _Both figures are hard to read._
>
> Thanks for bringing this up. **We will improve the readability of the figures.**
>
> > _Minor_
>
> Thank you for these comments. **We will clarify these points in the updated manuscript.**

---

> > ### Comment · Reviewer_KRhW · 2021-11-30
> > **Response to the revision**
> >
> > I would like to thank the authors for their valuable time. I can see that the authors have tried to address most, if not all, of the concerns raised by the reviewers in the revision. I have increased my score from 5 to 6 accordingly.
> >
> > I have one additional question: In the last sentence of Theorem 4.3, aren't the two $X_i$'s in the two occurrences of $\tilde{F}(X_i, \mathcal{S}_{n+1, j})$ typos? I pointed this out in my initial review, but the comment appears to have gone unnoticed. (This had no impact on my final score.)

---

> > > ### Author Response · Authors · 2021-11-30
> > > **Thank you**
> > >
> > > Thank you for taking the time to read our response and updated manuscript, and for the increase in score.
> > >
> > > Re: your additional question---yes, you are correct, this is a typo. Thank you for catching it! We apologize that we somehow missed this in our update, but will correct it in a future revision.

---

> ### Author Response · Authors · 2021-11-24
> **Response and updated manuscript**
>
> Dear Reviewer KRhW,
>
> Thank you once again for your time and helpful comments and suggestions. As posted in the general response, we have submitted a revised version of our paper that we believe addresses the questions that you raised. If you have any further comments, please do let us know.

---

### Official Review · Reviewer_2ZTR · 2021-11-03

**Correctness:** 1
**Technical Novelty And Significance:** 2
**Empirical Novelty And Significance:** 1
**Recommendation:** 6
**Confidence:** 4

**Main Review:**

**Strong points:**
* The framing of the problem is natural, and I like it (i.e., Sec 4.1 onwards).
* I like the idea of using the deep sets methodology.
* The three real-world data sets seem reasonably interesting / well-motivated.
* The paper is a pleasure to read.

**Weak points:**
* Experimental evaluation:
    * To me, the results are a little hard to interpret.  Your stated motivation (page 1, first paragraph) is to reduce the size of conformal prediction sets.  I may have missed it, but what are the sizes of the sets you get in Sec 5?  In particular, I think you should also report the conformal set sizes required to attain, say, 90% marginal coverage; otherwise, it's hard to say what exactly you are improving on.  It does seem though, from Figs 2,3, that to get 90% coverage on the in-silico example, you effectively need to include half the label set.  That's ... not great?
    * Along these lines, you seem to be "cheating" a little bit -- your stated motivation, i.e., cases when when the label space $\mathcal Y$ is large, seems to require ... $\mathcal Y$ to be large.  Yet you effectively truncate $\mathcal Y$ to 100 in all the experiments.  What happens if you don't do that?  Statistically?  Computationally?  I think it's OK to show when the method fails.
    * Also, it doesn't seem totally fair to not use CQC for the "Inner Sets" baseline, b/c then that baseline produces constant width intervals.  Meanwhile, yours can depend on $x$.

* Methodology: it seems the accuracy of deep sets as a conditional density estimator hinges on: (i) can $P_{Z \mid X}$ be decomposed in an additive way? and (ii) has the curse of dimensionality kicked in?  Accordingly, can you say how many dimensions there are in each of your examples?  Sorry if I missed it.

**Questions:** see my "Weak points" and "Additional feedback" sections.

**Additional feedback:**
* You seem to have switched between $S$ and $\mathcal S$ to denote a set on page 6.
* On page 6 -- "beach search" $\rightarrow$ "beam search".
* It's hard to tell what exactly is going on in the top row of Fig 2.
* Part (c) of Figs 2,3 -- TDR appears to asymptote at 0.8.  Why?
* I may have missed it, but how big is $\mathcal Z$ in each of the experiments?
* How do you set any tuning parameters for the deep sets neural network?  Doesn't this require additional data?
* I have what I think is a dumb question regarding the motivation of the paper.  Both your method and standard conformal will return a set.  But at the end of the day the practitioner still needs to use a single point to make a prediction.  How exactly does your method help with that problem?  I mean I get that you are trying to produce smaller sets, but you are still returning a set (i.e., more than one) of plausible values.

**Summary Of The Paper:**

**Summary:**
* The paper puts forth a greedy algorithm that maximizes the true discovery rate w.r.t. a set of labels (akin to coverage), subject to a user-specified restriction (i.e., constraint) on the false discovery rate, for the goal of building small but nevertheless valid prediction sets.
* A few calculations/results are worked out showing that the method works as advertised (i.e., TDR maximized, FDR controlled).
* Some empirical results are presented, where the method's TDR and FDR (along with those of one baseline from some other work) on three real-world data sets are evaluated.

**Summary Of The Review:**

**Recommendation:** reject.  I like the paper.  But I think there are some issues w/ the experimental evaluation that need to be sorted out before it gets accepted.  Happy to change my mind if I missed something.

---

> ### Author Response · Authors · 2021-11-12
> **Response (part 1)**
>
> We thank the reviewer for their comments and suggestions. We are committed to improving our paper and can easily address all concerns within the rebuttal period. Below, we address the reviewer’s comments and indicate in **bold** additional experiments or updates to the writing that we will post within several days. We hope that these will allay any concerns about our work and convince the reviewer that it will be a welcome contribution to the ICLR community. **If there are additional experiments or clarifications that we can perform to further support our case, please let us know.**
>
> > _Your stated motivation (page 1, first paragraph) is to reduce the size of conformal prediction sets. I may have missed it, but what are the sizes of the sets you get in Sec 5? In particular, I think you should also report the conformal set sizes required to attain, say, 90% marginal coverage; otherwise, it's hard to say what exactly you are improving on. It does seem though, from Figs 2,3, that to get 90% coverage on the in-silico example, you effectively need to include half the label set._
>
> Our goal is indeed to reduce the size of the conformal prediction sets, but only the noisy part. Since we are working in the generalized setting of multi-label classification, a prediction set may be large if either (1) there are many true positives, or (2) there are many false positives. The first is OK; the second we want to control---which we do in our paper. This is why we report the size of the false positive component of the set---both in Table 1 in terms of absolute numbers, and indirectly in Figs. 2 and 3 in terms of worst case FD violation.  **We agree that it would also be helpful to include the sizes of conformal sets under the standard calibration paradigm, and will include this for comparison.**
>
> With respect to the results on in-silico screening: it is a challenging task that modern methods still struggle on. The fact that it is challenging to recover all positives, even with large set sizes, emphasizes the need to control for the number of false positives (even at a cost of missing true positives), so that an experimental budget can be considered. We also emphasize that 100 is not the true upper bound on our label set: it is the truncation parameter that we set, as we state in the first paragraph of Sec. 5. We address this more directly in the next point.
>
> > _Along these lines, you seem to be "cheating" a little bit -- your stated motivation, i.e., cases when when the label space Y is large, seems to require … Y to be large. Yet you effectively truncate Y to 100 in all the experiments._
>
> There is a distinction between B, the hyper-parameter we set to manually truncate the label space, and $|\mathcal{Y}|$, the true size of the full label space. In standard conformal prediction, you are bound by the full size of the label set---in order to achieve valid coverage, you must consider all possible labels. In fact, this is the reason why output sets might become so large if your nonconformity measure is non-discriminative! Our procedure for truncating the label space $\mathcal{Y}$ is purely a design choice that is enabled by our new criterion. When we truncate $\mathcal{Y}$ to the top B individually ranked candidates (see Alg. 1 line 11) we are simply upper-bounding our false positive rate by B. This could be viewed as similar in spirit to a “hard” version of RAPS (Angelopoulous et. al., 2021c).
>
> We strongly emphasize that this is not “cheating”, but rather a _necessary_ component of our system. For example, if our true label space $\mathcal{Y}$ were extremely large or infinite (as it is in language generation tasks), then it would not be possible to achieve meaningful finite-sample expectation control (as discussed in remark 4.5). Making B smaller is also not without consequence. A lower B implies a lower upper bound on the TDR (we might cut out true positives due to truncation). Though it also makes $T_k$ less conservative (which could lead to higher empirical TDR). In our experiments, we simply choose B = 100 both for simplicity and efficiency, and for the purposes of visualization in Fig. 2 and 3; more generally, one could tune B on a development set.
>
> **We acknowledge and regret that this distinction is not clear in the current writing, and will update the manuscript accordingly (including stronger emphasis that B is a hyper-parameter, not an intrinsic quality of $\mathcal{Y}$, and a discussion of its impact). In particular, we will also correct the axes of Fig. 2 and 3 to be in terms of k/B, not k/|$\mathcal{Y}$| (a typo). We will encourage the reviewer to revisit this point after this update is posted.**

---

> > ### Author Response · Authors · 2021-11-12
> > **Response (part 2)**
> >
> > > _it seems the accuracy of deep sets as a conditional density estimator hinges on: (i) can P_{Z|X} be decomposed in an additive way? and (ii) has the curse of dimensionality kicked in? Accordingly, can you say how many dimensions there are in each of your examples? Sorry if I missed it._
> >
> > This is a good question. Our motivation for using deep sets as a conditional density estimator of $P_{Z|X}$ (EDIT: $P_{FP|X, \mathcal{S}}$), where $X$ is a set (EDIT: $\mathcal{S}$ is a set), comes from Thm. 2 of the deep sets paper, which states that a function $f(X)$ operating on a set $X$ is a valid set function iff it can be decomposed in the form $\rho(\sum_{x\in X} \phi(x))$ for suitable $\rho$ and $\phi$. Our input set $X$ (EDIT: $ \{ \phi(y_c, X) : y_c \in \mathcal{S} \} $) has elements of dimension 1, i.e., we directly use the raw individual label confidence (or you can think of it as the conformity score of the single label $y_c \in \mathcal{Y}$). Obviously, even though our deep sets model has the correct form of a valid set function, it still may be hard to learn the appropriately parameterized $\rho$ and $\phi$. It is easy, however, to plug in a better approximator if a practitioner should find one.
> >
> > > _TDR appears to asymptote at 0.8_
> >
> > In this case, about 20% of examples have no correct answers at all (i.e., some sentences do not contain any named entities). This is also true for the in-silico screening task as well. (And, it is also affected by the hard threshold B, as discussed above). This is another advantage of our framework over standard conformal prediction: requiring marginal coverage for inputs that may not have any correct answers is not well defined. In this case, our desired behavior does not change: we still want to control the total number of false positives.
> >
> > > _How do you set any tuning parameters for the deep sets neural network? Doesn't this require additional data?_
> >
> > We use some small amount of additional data, see line 6 of Alg. 1. The deep sets model is fast enough to train such that removing this small amount of data from the proper training set for the individual likelihood model doesn’t significantly affect the latter model’s accuracy. Note that this doesn’t need to be distinct data: as can sometimes be found in other works that use pipelined systems, we can also reuse the training set by splitting it into folds, training copies of the independent label predictor on k - 1 folds, and aggregating their (test) predictions on the k test folds (which we can use to train the deep sets model, albeit with some  train-test mismatch). **We will make this clearer in the manuscript.**
> >
> > > _Both your method and standard conformal will return a set. But at the end of the day the practitioner still needs to use a single point to make a prediction. How exactly does your method help with that problem?_
> >
> > This is a good point, and precisely the limitation of conformal prediction that we are trying to address! At the end of the day, for many problems, the output set must be validated. This can be expensive---see our illustration of this process in Fig. 1. In order to feasibly validate output sets (such as experimentally in a wet-lab), the number of false positives can’t be too large: otherwise, we have no hope of being able to verify valid individual predictions. This is what our work helps control.
> >
> > >  _I may have missed it, but how big is Z in each of the experiments?_
> >
> > Z is not a constant size, it varies per example. For example, it may be of size 0 (no correct answers), or even of size 50-70 (many correct answers). **We will add these dataset statistics to the appendix.**

---

> > ### Comment · Reviewer_2ZTR · 2021-11-29
> > **Response to the response**
> >
> > First of all, thanks for the additional work and spending the time to write up a response!
> >
> > * Re: inner sets -- I see.  OK, thanks for doing that.
> > * Re: $B$ -- OK, fair enough.  Thanks for explaining.
> > * Re: deep sets -- I'm a little confused.  I thought you use deep sets to model $P_{Z \mid X}$, where $Z$ is the response and $X$ is the feature vector.  So why does $X$ have dimension 1?  Is there a typo in the paper?  Are you trying to say you use deep sets to model the (unconditional) score distribution (which is indeed univariate)?
> >
> > In any event, I think I'll increase my score a little bit.

---

> > > ### Author Response · Authors · 2021-11-29
> > > **Reply**
> > >
> > > Thank you for taking the time to review our response, and for the increase in score.
> > >
> > > Re: $X$ -- we apologize, we misread your original question, and the notation used in our response is confusing (we were referring to a general $X$ that is an input to a set function, not necessarily the same $X$ in our paper). In the actual paper, $X$ is multidimensional (i.e., several hundred for image pixels or BERT/MPNN representations for text/SMILES sequences).
> > >
> > > The input to the set function (the DeepSets model) is a set of one dimensional features. The one dimensional features are derived from the multidimensional $X$ for a particular candidate $y_c$ using $\phi(x, y_c) = p_{\theta}(y_c \in Z  \mid x)$. The set function also doesn't directly model $P_{Z|X}$, but rather $P_{FP | X, \mathcal{S}}$ where $\mathcal{S} \subseteq \mathcal{Y}$ is the current set being considered, and $FP$ is it's number of false positives (an integer between $0$ and $|\mathcal{S}|$). Note that $FP$ is a (deterministic) function of $Z$ and the given $\mathcal{S}$, and so it's distribution is a pushforward of $P_{Z|X}$ (but we never explicitly model that distribution for our purposes).

---

> ### Author Response · Authors · 2021-11-24
> **Response and updated manuscript**
>
> Dear Reviewer 2ZTR,
>
> Thank you once again for your time and helpful comments and suggestions. As posted in the general response, we have submitted a revised version of our paper that we believe addresses the questions that you raised. If you have any further comments, please do let us know.

---

### Author Response · Authors · 2021-11-18
**Updated Manuscript**

To our reviewers,

Thank you for your helpful comments and suggestions. We have addressed many of the stated concerns in our now updated draft. We hope that these clarifications will allay any concerns about our work and convince the reviewers that it will be a welcome contribution to the ICLR community. Once again, if there are additional experiments or clarifications that we can perform to further support our case, please let us know.

We have also included as supplementary material a version of our paper with major changes highlighted in red to help the reviewers navigate the updates.

To summarize our main changes:

1. We added an extensive clarification as to the role of the hyper-parameter B in truncating the examined label space and what effects it has. The reviewers were correct in flagging this point; we hope and expect that it is now significantly clearer.

2. We have added additional sections to the Appendix, including extra experimental results, dataset details, and commentary on practical considerations (k-FWER vs FDR, how to set k, advantages and failures of the greedy set construction, our motivation for DeepSets, etc.).

3. We have updated Algorithm 1 to make it more precise and abstract, while also indicating where our proposed (implemented) models fit in.

4. We have added a comparison to standard coverage preserving conformal prediction (at level 90). Additionally, we implemented CQC for the inner and outer sets methods---however, while doing validation, though we found that the worst case coverage violations was better (i.e., indicating better conditional coverage), the TPR was slightly worse than before in some cases. Since this approach (inner sets) is already quite conservative, worst case violation is really not a problem, and rather TPR is the more telling metric to maximize, so we kept our previous results (the ones which do not use CQC). We added a note explaining this in our manuscript.

5. There has been extensive discussion between comparing FDR and our method. We address this in Section 2, as well as in Appendix D. We also realize that our usage of “false discovery” is perhaps confusing with that in FDR, and have more explicitly switched our terminology to simply refer to “false positives” (our prior usage of false discovery was motivated by our applications in drug discovery). We have updated this in all definitions, discussion, and title.

6. We have corrected the typos that reviewers have pointed out, and have updated much of our indexing notation. In addition, we now use upper-case Z to refer to the random variable, and lower-case z (as opposed to $\mathcal{Z}$) to refer to its value.

7. We have updated the readability and scales of Figures 2 and 3. We were also to catch a reporting error for the inner sets method in Figure 2 (a), which had caused the scale to be off.

We thank the reviewers again for their time, and look forward to hearing additional feedback.

---

### Decision · Program_Chairs · 2022-01-20

**Decision:**

Reject

**Comment:**

This paper suggests that in multi-label classification (where there are multiple y that could be correct), the usual conformal prediction setup could be too conservative (too large a set with too many false positives), because it asks for "full" coverage. They propose to change the error metric from coverage to precision, to instead output a smaller set, with higher precision, potentially with the loss of coverage. The paper thus produces two variants of conformal prediction: guaranteeing that the expected number of false positives is at most k, and that the probability that the #false positives being > k is < \delta. The experimental study is interesting.

I followed the extensive discussion thread, and appreciate the authors' and reviewers' willingness to engage. However, in the end, the (excellent) reviewers were somewhat still not enthusiastic about the paper, and with nobody willing to champion the paper, it ended up on the borderline, in the bottom half of my set. Nevertheless, I read through the paper in detail myself to make sure, but I find enough reasons that suggest that the paper is not ready for publication currently, and would benefit from a significant overhaul.

It seems like the final algorithm is a slight variant of nested conformal prediction (a well known and oft-cited paper by Gupta et al, 2020 that the authors seem to miss) in the following sense. The usual conformal-for-classification framework would order the labels in terms of a score (like posterior probability) and then return a set of labels whose score is less than some threshold. (The same style of procedure can be used for the single-label and multi-label case also.) The nesting appears to be the same high level framework used in this paper, except that the threshold is chosen by a different rule from standard conformal prediction (ie same nesting, different threshold). Writing the algorithm more transparently will make for a tighter connection to the conformal literature --- at the moment, the authors claim a fair bit of novelty, but this is partially due to the omission of this reference and the cleaner broad (nested conformal) framework under which this work (as well as standard conformal) sit.

At the same time, I was not fully convinced of the central theoretical claim of the paper, in Proposition 4.6. The authors claim that since Theorem 4.3 is simultaneously valid for all j in B satisfying a condition (the filtered set), the theorem can also be invoked for a data-dependent index (chosen via (12)). This does not appear to the true, and at the very least requires careful justification. At a high level, dropping the conditioning for simplicity, Thm 4.3 reads as "forall j in [B], E[A_j] <= c", but this does not imply that for a data-dependent j-hat, we have E[A_{j-hat}] <= c. It would have been true, if the forall j appeared as a sup_j inside the expectation (or as a forall j inside the probability, rather than outside). The authors may want to clarify more carefully, if it is indeed true.

Minor: I also continue to find typos in the main results and proofs. These are minor, but should be corrected. I believe that in the last line of Theorem 4.3, the X_i should be X_{n+1}. In the display after (25), that should be T_k, and not T_{k,\delta}. There appear to be some other potentially missing references as well. For example, while distribution-free conformal approaches are cited, distribution-free calibration approaches are not, within the related work section. At the same time, some very recent papers, such as by Bates et al, or Angelopoulos et al, appear to be overemphasized in the introduction. A more fair coverage of related work could be useful.